# SPADA: A Verifiable Test-Driven Agent for Controllable Parametric CAD Assembly Generation

Keyou Zheng [1 2]  Xuyang Su [1 2]  Jiewu Leng [1 2]

## Abstract

Parametric CAD is widely used in mechanical and product engineering, but current generative models still struggle to produce assemblies that are both editable at the parameter level and consistent with inter-part constraints. Mesh and history-free B-rep methods represent multi-part shape but often lack the flexible structure and constraint logic needed for reliable downstream edits; code-based CAD generation offers direct parametric control, yet most settings focus on single-part solids rather than constrained assemblies. We introduce SPADA (Self-testing Parametric Assembly Design Agent), a test-driven agent that synthesizes assembly code together with deterministic verification tests, and uses these tests as an executable contract for controllable generation. SPADA runs an iterative compile–test–repair loop with multimodal feedback, checking both specification logic and physical feasibility through programmatic constraints. To support evaluation, we release SPADA-Bench-Verified, a human-verified benchmark of real-world code-centric assemblies paired with deterministic tests including engineering constraints. Experiments show that SPADA produces complex assemblies while maintaining geometric fidelity, supporting test-driven agents as a concrete path toward reliable, controllable CAD generation.

## 1. Introduction

Computer-aided design (CAD) underpins modern engineering and manufacturing, transforming conceptual designs into precise 3D geometry for simulation and manufacturing processes (Shah & Mäntylä, 1995). Contemporary CAD is predominantly *parametric*: models are defined by editable parameters, reusable features (Camba et al., 2016), and explicit relations (*e.g.*, dimensions, alignments, mates), allowing engineers to iterate without rebuilding geometry from scratch. Despite the ubiquity of CAD software, the design process remains a labor-intensive bottleneck, requiring substantial human expertise to translate abstract requirements into valid, constrained 3D assemblies (Rankohi et al., 2022). Recent work addresses this bottleneck through several generative paradigms. Sequence-based methods predict modeling histories (Wu et al., 2021; Willis et al., 2021), while B-rep approaches synthesize topology directly (Lambourne et al., 2021; Xu et al., 2024). These methods generate single-part shapes but often produce static geometries lacking the structured code logic needed for downstream editing.

Code-as-a-Model, or **code-based CAD generation**, offers a promising representation. As Fig. 1 shows, executable CAD programs (in CADQuery [1] or OpenSCAD [2]) expose *white-box* editability and leverage large language models (LLMs) pre-trained on code corpora (Govindarajan et al., 2026). Unlike graphical systems that store designs as opaque feature trees, code-based CAD leverages standard programming constructs. Programmatic representation supports automated testing and modular reuse, making it an ideal substrate for generative design (Kolodiazhnyi et al., 2025). Code-based CAD exposes assembly structure through explicit calling and module hierarchies, enabling precise specification of relational constraints between components.

However, a critical gap remains. Current code-based methods focus on single-part solids and treat syntactic correctness or visual similarity as proxies for quality, failing to address the core complexity of engineering design, namely assemblies. Assembly constraints have been studied via B-rep representations (Jones et al., 2021; Willis et al., 2022), but these methods do not address code-based parametric generation where multiple parts must satisfy precise constraints. A code that compiles is insufficient if parts collide or fail to fit.

---

[1]Guangdong Provincial Key Laboratory of Computer Integrated Manufacturing, Guangdong University of Technology [2]State Key Laboratory of Precision Electronic Manufacturing Technology and Equipment, Guangdong University of Technology. Correspondence to: Jiewu Leng <jwleng@gdut.edu.cn>.

*Proceedings of the 43rd International Conference on Machine Learning*, Seoul, South Korea. PMLR 306, 2026. Copyright 2026 by the author(s).

[1]`https://github.com/CadQuery/cadquery`
[2]`https://openscad.org`

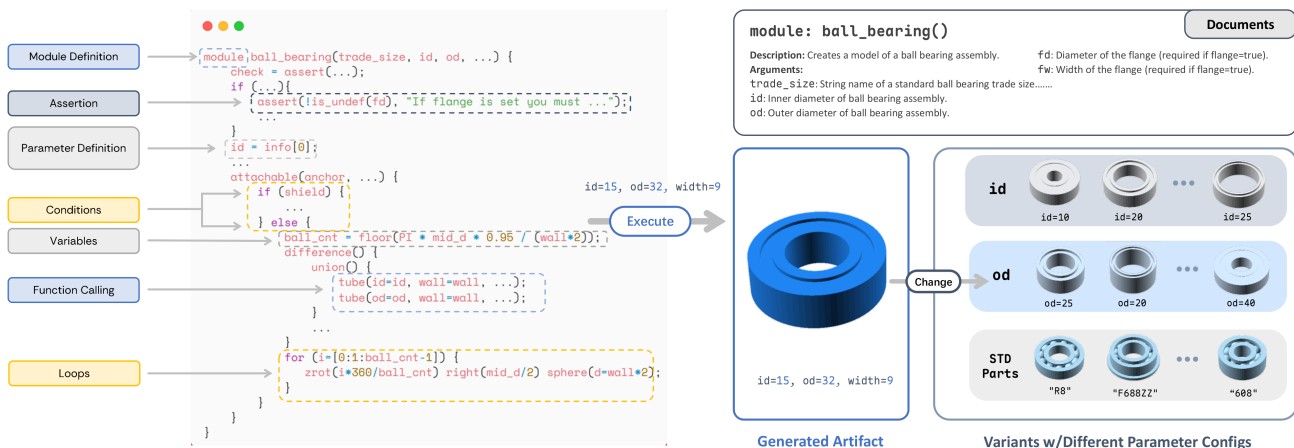

*Figure 1.* **Code-as-a-Model Paradigm.** Code-based CAD defines geometry through executable logic rather than static geometric records such as meshes or B-reps. Programming constructs such as assertions, loops, and parameters directly determine geometry, allowing a single source module to generate design families by changing values such as `id` and `od`.

To bridge this gap, we treat CAD generation as test-driven development (TDD) rather than simple translation. Test execution lets an agent form hypotheses, validate them, and repair designs against an explicit contract. We introduce SPADA (Self-testing Parametric Assembly Design Agent), an agentic framework that generates parametric assembly code alongside deterministic verification tests. SPADA compiles code, inspects geometry with mixed feedback, and enforces engineering constraints through tests, moving beyond syntax-level correctness. We also curate SPADA-Bench-Verified, a benchmark of diverse code-centric CAD tasks with deterministic constraint verifiers.

In summary, our contributions are as follows:

1. We develop SPADA, a verifier-guided agent that employs an iterative compile-test-repair loop with rich environmental feedback to satisfy both geometric fidelity and constraint satisfaction.

2. We introduce SPADA-Bench-Verified, a challenging benchmark for code-based parametric CAD assemblies with verifiable constraints, shifting evaluation from geometry similarity to *fine-grained constraint verification*.

3. Experiments demonstrate that SPADA outperforms baselines in generating valid, controllable parts and assemblies while maintaining high code quality.

## 2. Related Work

**Parametric CAD Representation and Generation.** CAD models are represented via boundary representation (B-rep), sketch-extrude sequences, or program code. B-rep generative methods include BrepNet (Lambourne et al., 2021),

UV-Net (Jayaraman et al., 2021), and diffusion-based approaches (Xu et al., 2024; Jayaraman et al., 2023; Lee et al., 2025; Guo et al., 2025; Xu et al., 2025; Liu et al., 2025; Li et al., 2025a). Sequence-based methods mirror human design history; DeepCAD (Wu et al., 2021) and Fusion 360 Gallery (Willis et al., 2021) provide foundational datasets, while recent works extend this paradigm to image-conditioned generation (Alam & Ahmed, 2025; Li et al., 2025c; Khan et al., 2024a; Rukhovich et al., 2025). Sketch generation methods (Para et al., 2021; Li et al., 2020; Chereddy & Femiani, 2025; Wu et al., 2025) target specialized representations. Research on *assemblies* remains limited; JoinABLe (Willis et al., 2022) and AutoMate (Jones et al., 2021) predict joints for existing geometry. In contrast, SPADA-Bench-Verified comprises *code-based parametric workspaces* with explicit parameters and part reuse, enabling relational constraints to be satisfied from scratch.

**LLMs for Code-based CAD.** LLMs have demonstrated proficiency in program synthesis (Chen et al., 2021), prompting test-driven evaluation frameworks such as SWE-bench (Jimenez et al., 2024). In CAD, fine-tuning approaches like CADmium (Govindarajan et al., 2026) and Don't Mesh with Me (Mews et al., 2025) target sequential design; CAD-Coder (Doris et al., 2025) and CADCode-Verify (Alrashedy et al., 2025) explore code-execution-refinement loops. Multimodal systems including CAD-Assistant (Mallis et al., 2025) and Cadrille (Kolodiazhnyi et al., 2025) leverage vision-language capabilities, while CADTalk (Yuan et al., 2024) and CADReview (Chen et al., 2025) address semantic understanding and error repair. Prior work emphasizes single-part generation evaluated via geometric similarity; SPADA advances this by adopting a verifier-driven framework where relational constraints are

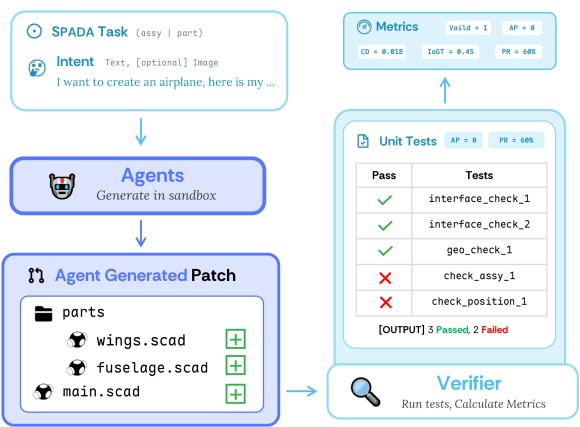

*Figure 2.* **SPADA-Bench-Verified Task Execution.** Given a task intent, the sandboxed agent generates a multi-file CAD patch. The verifier executes the workspace, runs unit tests, and reports geometric metrics such as PR and IoU.

first-class, automatically testable requirements.

**CAD Generation with Self-Refinement.** Self-refinement iterates between generation and feedback using execution signals, as exemplified by ReAct (Yao et al., 2023) and Reflexion (Shinn et al., 2023). In CAD, CADCodeVerify (Al-rashedy et al., 2025) introduces verification mechanisms; Seek-CAD (Li et al., 2025b) and EvoCAD (Preintner et al., 2025) employ feedback-guided optimization; GenCAD-Self-Repairing (Tsuji et al., 2025) repairs invalid construction sequences. Visual-only verification requires VLMs to infer measurements from pixels, introducing scale and orientation errors; subtle violations remain invisible, and judgments vary non-deterministically. SPADA addresses *program-level* refinement (compilation and manifold geometry) and *assembly-level* refinement (inter-part constraints), shifting pass/fail decisions from non-deterministic VLM judgments to deterministic executable predicates.

## 3. SPADA-Bench-Verified

Inspired by SWE-bench (Jimenez et al., 2024), we introduce SPADA-Bench-Verified, a **verifiable, execution-based** benchmark comprising **10k single-part** and **1.7k assembly** tasks derived from high-quality human code. It spans three code-based CAD ecosystems (OpenSCAD, CadQuery, Build123d) and pairs each task with deterministic verifiers for objective pass/fail evaluation.

### 3.1. Task Definition

Evaluating generative CAD models is difficult because visual similarity does not ensure editability, valid geometry, or assembly-level constraint satisfaction. Existing datasets also lack the multi-file workspaces needed to test modern CAD agents.

Each benchmark task provides a natural-language intent $\mathcal{I}$, a code workspace with a designated entry point $e$, and a deterministic verifier suite. We evaluate two task families through the execution flow in Fig. 2:

- **Editable Part Generation**: A single-part task has $|\mathcal{P}| = 1$ and an interface contract $\Gamma$ describing the exported symbol and parameters. The agent must produce editable parametric code, not a static mesh, and the generated part must execute without external dependencies while satisfying geometry constraints.
- **Assembly Generation**: An assembly task has $|\mathcal{P}| > 1$ and a constraint set $\mathcal{C}$ grounded in instance names and anchors. The agent must coordinate multiple files, instantiate valid parts, and apply transforms or joints that satisfy relational constraints such as mating and collision avoidance.

**Tests.** Given a candidate workspace $\mathcal{W}$, the harness executes $e$ in a controlled environment and runs unit tests $\tau \in \mathcal{T}$ for interface compliance, geometric validity, and constraint satisfaction.

**Metrics.** We define two levels of evaluation metrics. At the *test level*, for any test $\tau$, we define its outcome on $\mathcal{W}$ as $r_\tau(\mathcal{W}) = 1$ if passed, and 0 otherwise. The **Pass Rate (PR)** measures the mean test pass rate *within a single task*:

$$\text{PR}(\mathcal{W}) = \frac{1}{|\mathcal{T}|} \sum_{\tau \in \mathcal{T}} r_\tau(\mathcal{W}). \tag{1}$$

At the *task level*, we define strict task pass using an execution indicator $E(\mathcal{W}) \in \{0, 1\}$:

$$\text{TP}(\mathcal{W}) = \begin{cases} 1, & E(\mathcal{W}) = 1 \ \wedge \ \text{PR}(\mathcal{W}) = 1, \\ 0, & \text{otherwise}. \end{cases} \tag{2}$$

To evaluate complete task success, we introduce **All Pass Rate (APR)**, the fraction of tasks where *all* tests pass, computed as the dataset-level mean of $\text{TP}(\mathcal{W})$ across tasks. It measures the agent's ability to satisfy all design constraints simultaneously:

$$\text{APR} = \frac{1}{N} \sum_{i=1}^{N} \text{TP}(\mathcal{W}_i), \tag{3}$$

where $N$ is the number of tasks and $\mathcal{W}_i$ is the candidate workspace for task $i$.

### 3.2. Benchmark Construction

**Data processing pipeline.** We construct SPADA-Bench-Verified from a large corpus pool of real CAD programs through a multi-stage pipeline. We preprocess candidate

workspaces with a shared normalization pipeline and execute each candidate, retaining only those that run successfully and produce valid geometry. To reduce training overlap, we rewrite task descriptions into a unified intent template using an LLM. From the executable pool, we derive both task families by extracting entry points, interfaces, and constraints. Based on these, we generate deterministic verifiers using LLM-assisted templates, followed by human verification to ensure correctness and coverage. For further details, see Appendix C.

**Code-based execution environment.** We evaluate candidates by running their workspace inside a sandbox with pinned toolchain versions. The harness executes the designated entry point and collects structured feedback: execution status, logs, and produced artifacts. If execution fails, $E(\mathcal{W}) = 0$ and the strict task pass becomes 0. If successful, tests run against the realized outputs using the golden reference settings.

**Deterministic verifiers.** Each task includes a small set of benchmark-provided deterministic verifiers that act as an executable contract for the intended design. These verifiers check that the submitted workspace exposes the required parameter interface, produces valid geometry, remains stable under mild parameter changes, and (for assemblies) satisfies explicit relational requirements such as correct part instantiation, non-interference, and basic mating. Because the verifiers are deterministic, the same submission yields the same outcome across runs, removing ambiguity from purely visual inspection. We audit verifier quality on 200 sampled tasks and report per-language and difficulty breakdowns in Appendices C.3 and C.6.

### 3.3. Dataset Statistics

Table 1 compares SPADA-Bench-Verified with representative human-written code datasets (Yuan et al., 2024; Alrashedy et al., 2025; Chen et al., 2025). We focus on three essential properties of code-based CAD: editability, which is critical for parametric design; readability, which affects maintainability; and human-likeness, which reflects whether generated code aligns with human practice. To evaluate these properties efficiently at scale, we extend the LLM-as-a-judge scoring protocol used in CADmium (Govindarajan et al., 2026). The rubrics and further dataset details are provided in Appendix C.4.

## 4. SPADA: Self-testing Parametric Assembly Design Agent

We present SPADA (Self-testing Parametric Assembly Design Agent), an LLM-driven agent for controllable parametric assembly generation through test-driven development.

*Table 1.* **Comparison to human CAD datasets.** Assy indicates assembly-level tasks. Edit, Read, Hum are LLM-judge scores (0 to 5) for editability, readability, and human-likeness. Best in **bold**, second-best underlined.

| Dataset | CAD Language | | | Assy | Edit | Read | Hum | #Tasks |
|---|---|---|---|---|---|---|---|---|
| | SCAD | CQ | B123d | | | | | |
| CADTalk-Real (Yuan et al., 2024) | ✓ | – | – | – | 3.45 | 3.85 | 3.89 | 45 |
| CADPrompt (Alrashedy et al., 2025) | – | ✓ | – | – | 3.88 | **4.32** | **4.18** | 200 |
| CADReview (Chen et al., 2025) | ✓ | – | – | – | 3.78 | 3.48 | 3.57 | 1.5k |
| SPADA (Ours) | ✓ | ✓ | ✓ | ✓ | **4.35** | 4.27 | 4.14 | **11.7k** |

*Table 2.* **SPADA Tool Suite.** The complete tool suite combines two general tools with three CAD-specific tools. Full specifications appear in Appendix A.

| Tool | Function |
|---|---|
| `file_editor` | Create/edit workspace files |
| `terminal` | Execute allowed shell commands |
| `spec` | $\mathcal{I} \to (\Theta, \Gamma, \mathcal{C})$: extract constraints |
| `inspect` | $\mathcal{W} \to (s, \mathcal{G}, \mathcal{V})$: compile and observe |
| `verifier` | $(\mathcal{W}, \text{tests}) \to \mathcal{R}$: execute tests |

SPADA employs a minimal yet effective tool suite: beyond standard file editing and terminal access, only three CAD-specific tools (`spec`, `inspect`, and `verifier`) drive the entire workflow (Table 2).

### 4.1. Framework and Algorithm

**Formalization.** We formalize SPADA as an agent $\mathcal{A}$ interacting with an execution environment $\mathcal{E}$ through a fixed tool interface $\mathcal{T} = \{\mathcal{T}_i\}_{i=1}^N$. The agent $\mathcal{A}$, implemented by a multimodal LLM, maintains context $c_t$ comprising the original intent, workspace state, and accumulated feedback. The environment $\mathcal{E}$ provides a sandboxed harness with pinned toolchain versions that executes CAD code and returns structured observations. Given intent $\mathcal{I}$, at timestep $t$ the agent generates plan $p_t$ and action $a_t$:

$$p_t \leftarrow \mathcal{A}(\mathcal{I}; c_{t-1}, \mathcal{T}), \ a_t \leftarrow \mathcal{A}(p_t; c_{t-1}, \mathcal{I}, \mathcal{T}). \quad (4)$$

Executing $a_t$ produces feedback $f_t$ and state $e_t$: $(f_t, e_t) \leftarrow \mathcal{E}(a_t; e_{t-1}, \mathcal{T}, \mathcal{I})$. The context updates as $c_{t+1} \leftarrow \text{concat}(f_t, \{c_s\}_{s=1}^t)$, and the loop continues until all tests pass or the budget is exhausted.

**Main Loop.** Algorithm 1 details the main loop. The agent first calls `spec` to convert natural-language intent into a structured specification containing parameters $\Theta$, interface contracts $\Gamma$, and constraint predicates $\mathcal{C}$. It then generates an initial workspace and enters the iterative loop: `inspect` compiles the code and returns execution status, geometric metadata (bounding box, volume, manifold status), and

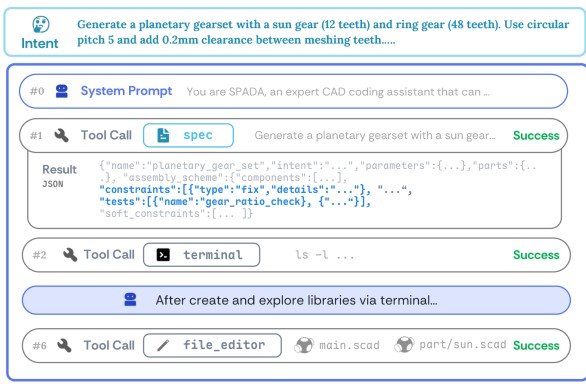
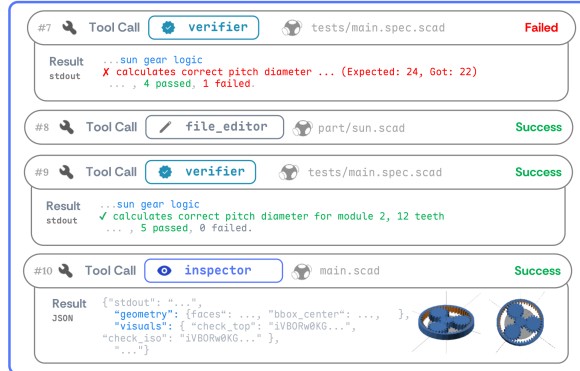

*Figure 3.* **Test-Driven Agent Trajectory.** The `verifier` provides quantitative feedback (reporting exact discrepancies between expected and measured values) that enables mathematically precise corrections. Here, the verifier detects an incorrect pitch diameter (b), and the agent computes the corrective translation to achieve convergence.

optional multi-view renders. If compilation fails, the agent repairs syntax errors and continues; otherwise, it generates deterministic tests from the specification and runs `verifier`, which returns structured diagnostics including measured values, expected bounds, and code locations. This feedback enables targeted patches until all tests pass or iterations are exhausted.

---

**Algorithm 1** SPADA Main Loop

---

**Require:** Intent $\mathcal{I}$, max iterations $T$
**Ensure:** Workspace $\mathcal{W}$, report $\mathcal{R}$
1: $\text{spec\_data} \leftarrow \text{spec}(\mathcal{I}); \mathcal{W} \leftarrow \text{generate}(\text{spec\_data})$
2: **for** $t = 1$ **to** $T$ **do**
3:     $\text{obs} \leftarrow \text{inspect}(\mathcal{W})$
4:     **if** $\text{obs.status} = \text{error}$ **then**
5:         $\mathcal{W} \leftarrow \text{repair}(\mathcal{W}, \text{obs});$ **continue**
6:     **end if**
7:     $\text{tests} \leftarrow \text{gen\_tests}(\text{spec\_data}, \text{obs})$
8:     $\mathcal{R} \leftarrow \text{verifier}(\mathcal{W}, \text{tests})$
9:     **if** $\mathcal{R}.\text{all\_passed}$ **then**
10:         **return** $\mathcal{W}, \mathcal{R}$
11:     **end if**
12:     $\mathcal{W} \leftarrow \text{apply\_patches}(\mathcal{W}, \text{plan\_repairs}(\mathcal{R}, \text{obs}))$
13: **end for**
14: **return** $\mathcal{W}, \mathcal{R}$

---

### 4.2. Deterministic Verification

**Test Generation.** Unlike general code synthesis where tests are given, CAD assemblies require *generating* test code that encodes geometric constraints. By leveraging standard test frameworks (e.g., pytest), SPADA synthesizes tests directly from the extracted specification: each constraint $c_k \in \mathcal{C}$ compiles into a test function $\tau_k : \mathcal{G} \to \{\text{pass}, \text{fail}\}$. Verification becomes $\text{Verify}(\mathcal{W}) = \prod_{k=1}^{K} \mathbf{1}[\tau_k(\text{exec}(\mathcal{W})) = \text{pass}]$. The verifier returns structured reports $\mathcal{R} = \{(k, r_k, m_k, e_k, \ell_k)\}_{k=1}^{K}$, where $r_k$ is the pass/fail result, $m_k$ the measured value, $e_k$ the expected value, and $\ell_k$ the relevant code location. This is semantic output verification rather than syntax checking; Appendix C.3 details the distinction.

**Supporting Tools.** The supporting tools (`spec` and `inspect`) feed the verifier with structured information. The `spec` tool converts ambiguous natural language into explicit constraint predicates, extracting quantitative parameters and relational constraints while enabling user review to catch misinterpretations before generation. The `inspect` tool combines execution feedback (logs, errors) with visual feedback (multi-view renders) and part-level introspection, addressing limitations of either modality alone: visual-only approaches suffer VLM pixel inference errors, while execution-only approaches lack geometric semantics. Implementation details of tool interfaces, test templates, and the execution harness are provided in Appendix A.

**Illustrative Example.** Figure 3 details an execution trace for a planetary gear assembly task, demonstrating the efficacy of SPADA's test-driven loop. The agent begins by using the `spec` tool to translate the natural language prompt into a structured contract, defining critical parameters (e.g., *circular pitch 5*, *0.2mm clearance*) and success criteria. A critical intervention occurs at **Step #7**: while the generated code compiles, the `verifier` detects a constraint error: the calculated pitch diameter is 22mm, deviating from the required 24mm. Unlike visual inspection, which fails to identify such subtle dimensional inaccuracies, the verifier provides a deterministic assertion failure. With the quantitative feedback, the agent interprets the log to identify the flaw in the gear logic, executes a precise `file_editor` action (Step #8) to correct the module calculation, and achieves a passing test suite (Step #9) and a manufacturable assembly (Step #10). This trajectory underscores the necessity of execution-based feedback: without the explicit signal from the verifier, the model would likely hallucinate a visually plausible but mechanically invalid gearset.

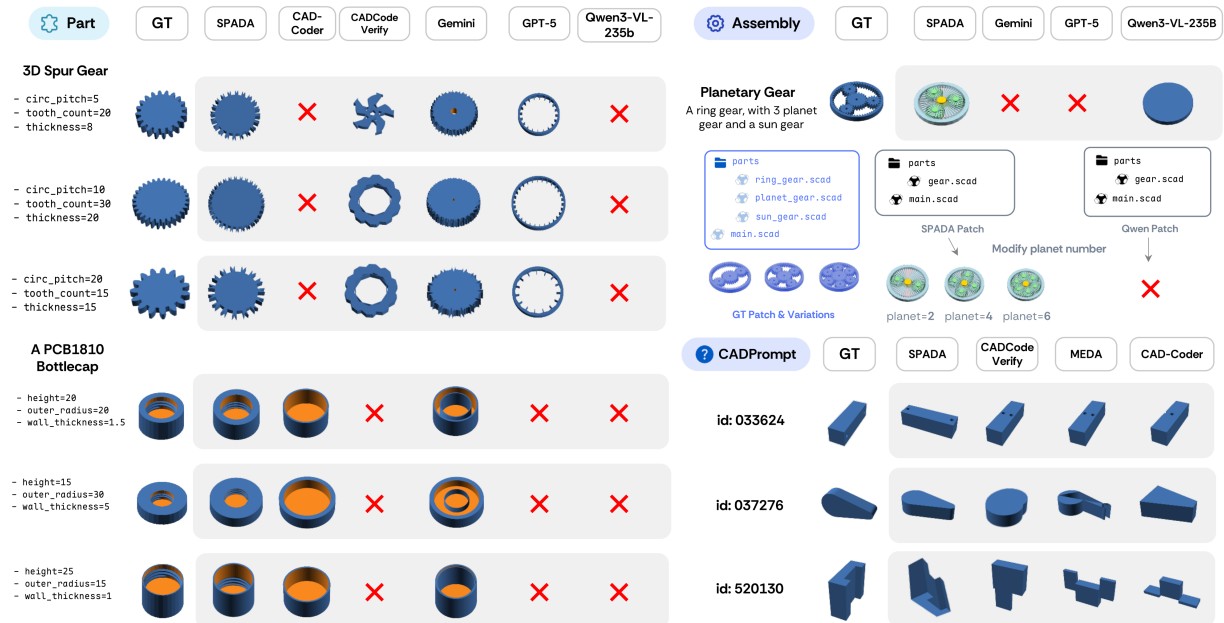

*Figure 4.* **Qualitative Comparison.** Baseline models produce visually plausible but functionally invalid geometry (*e.g.*, misaligned gears, incorrect thread pitches). SPADA uses its verification loop to satisfy strict engineering constraints, ensuring manufacturability and correct assembly alignment (Section 5.2).

# 5. Experiments

## 5.1. Experimental Setup

**Datasets.** We evaluate primarily on SPADA-Bench-Verified (Section 3). Because CADQuery has fewer public human-written parts and assemblies than OpenSCAD, we also report results on CADPrompt (Alrashedy et al., 2025), a widely adopted human-code benchmark for text-to-CAD generation.

**Baselines.** We compare SPADA against three types of baselines. For SPADA-Bench-Verified parts and CADPrompt tasks, we include representative code-based CAD generation methods such as CAD-Coder (Doris et al., 2025) and MEDA (Panta et al., 2025), the self-refinement method CADCodeVerify (Alrashedy et al., 2025), and zero-shot LLMs and VLMs. For SPADA-Bench-Verified assembly tasks, since no prior methods address workspace-level CAD code generation, we evaluate open-weight and closed-source multimodal LLMs using the same prompting as SPADA.

**Models.** We evaluate multiple multimodal LLMs and VLMs supporting tool calls, including open-weight models (Qwen3-VL-30B-A3B, Qwen3-VL-235B-A22B (Bai et al., 2025)) and closed-source models (OpenAI GPT-4.1, GPT-5, Gemini-3-flash-preview). All models are accessed via their official APIs. Unless otherwise specified, SPADA uses Gemini-3-flash-preview as the backbone model. Detailed model configurations and prompting strategies are provided in Appendix A.3.

**Metrics.** Following CAD generation evaluation protocols (Khan et al., 2024b; Alrashedy et al., 2025), we assess three aspects when evaluating tasks, namely validity, geometric fidelity, and constraint satisfaction.

For *validity*, we report **Invalid Rate (IR)**, the percentage of scripts that fail to compile or produce invalid geometry. Lower IR indicates more reliable code generation, though low IR alone does not guarantee correct geometry.

For *geometric fidelity*, we adopt point cloud-based metrics. **Chamfer Distance (CD)** measures bidirectional average nearest-neighbor distance between generated and ground truth point clouds, where lower values indicate better shape matching. **Hausdorff Distance (HDD)** captures worst-case deviation and is more sensitive to outliers. Both are reported as mean(median) scaled by $10^{-3}$. **Intersection over Union (IoU)** measures volumetric overlap via voxelization, where higher values indicate better coverage.

For *constraint satisfaction*, we report **Pass Rate (PR)**, the fraction of all unit tests passed across all test executions within a task, and **All Pass Rate (APR)**, the fraction of tasks where all verifiers pass in a single execution. APR is the strictest metric, requiring valid geometry, correct interfaces, tolerances, and parametric robustness simultaneously. Full

*Table 3.* **Part generation results on SPADA-Bench-Verified.** CD is scaled by $10^{-3}$. Best in **bold**, second-best underlined.

| Method | IR (%)↓ | IoU (%)↑ | CD↓ | PR (%)↑ | APR (%)↑ |
|---|---|---|---|---|---|
| *Zero-shot LLMs & VLMs* | | | | | |
| Qwen3-VL-30B-A3B | 31.2 | 32.8 | 158.4 (134.7) | 10.7 | 4.2 |
| Qwen3-VL-235B-A22B | 19.5 | 42.3 | 108.8 (102.5) | 13.2 | 6.9 |
| GPT-4.1 | 22.2 | 43.7 | 128.9 (112.8) | 14.4 | 10.8 |
| GPT-5 | 4.2 | 66.8 | 28.7 (24.3) | 40.8 | 29.7 |
| Gemini-3-flash-preview | 3.1 | 61.2 | 27.5 (24.8) | 46.5 | 35.2 |
| *Text-to-CAD Methods* | | | | | |
| CAD-Coder (Doris et al., 2025) | 12.8 | 48.3 | 38.9 (34.2) | 18.5 | 10.3 |
| CADCodeVerify (Alrashedy et al., 2025) | 10.4 | 54.6 | 31.5 (27.8) | 24.7 | 15.8 |
| **SPADA (Ours)** | **1.8** | **78.9** | **11.2 (10.8)** | **65.7** | **52.3** |

metric definitions are in Appendix C.5.

## 5.2. Main Results

**Part Generation.** Table 3 compares SPADA against text-to-CAD methods and frontier multimodal models on SPADA-Bench-Verified. Zero-shot models struggle with precise parametric logic, with smaller models exhibiting invalid rates above 19% due to syntax errors and API misuse. Frontier models such as GPT-5 and Gemini-3-flash achieve reasonable validity (3–4% IR) but still fail on constraint satisfaction, with APR remaining below 36%. SPADA reduces IR to 1.8% and improves APR to 52.3%, demonstrating that verifier-guided iteration substantially improves both code correctness and constraint satisfaction. On geometric fidelity, SPADA achieves 78.9% IoU and CD of 11.2 (10.8), outperforming baselines. Remaining failures stem from complex parametric dependencies that require multi-step geometric reasoning beyond current model capabilities.

**Assembly Generation.** Assembly tasks pose additional challenges beyond single-part generation (Table 4). Baseline models frequently fail due to hallucinations involving outdated or deprecated package APIs. These errors persist even after multiple self-correction attempts without execution feedback. SPADA mitigates this by capturing deprecation warnings and runtime errors from the execution environment, enabling targeted repairs. The verifier-guided loop improves APR from 20.4% (best zero-shot) to 41.9%, though absolute performance indicates that multi-part spatial reasoning remains challenging for current LLMs.

**Qualitative Analysis.** Figure 4 illustrates the qualitative differences between SPADA and baseline approaches. In the **3D Spur Gear** and **Bottlecap** tasks (left), baseline models frequently hallucinate geometry that appears plausible but fails dimensional checks, such as incorrect tooth counts or solid shells lacking internal voids. In the **Planetary Gear** assembly (right), purely visual generation leads to component collisions and invalid gear ratios. SPADA, by contrast, uses `verifier` feedback to correct these parameters iter-

*Table 4.* **Assembly results on SPADA-Bench-Verified.** CD is scaled by $10^{-3}$. Best in **bold**, second-best underlined.

| Method | IR (%)↓ | CD↓ | PR (%)↑ | APR (%)↑ |
|---|---|---|---|---|
| Qwen3-VL-30B-A3B | 38.6 | 142.3 (118.5) | 7.3 | 2.5 |
| Qwen3-VL-235B-A22B | 26.9 | 56.7 (48.2) | 12.8 | 5.4 |
| GPT-4.1 | 21.3 | 89.4 (76.9) | 17.5 | 8.2 |
| GPT-5 | 8.7 | 35.8 (30.5) | 26.3 | 14.7 |
| Gemini-3-flash-preview | 5.4 | 22.4 (19.1) | 32.8 | 20.4 |
| **SPADA (Ours)** | **2.4** | **1.8 (1.5)** | **54.2** | **41.9** |

atively, ensuring that the final output not only resembles the ground truth but functions correctly as a mechanical assembly. In the **CADPrompt** tasks (bottom), all methods achieve reasonable geometric fidelity, but subtle differences in edge chamfers and hole placements distinguish constraint-verified outputs from approximations.

**Generalization on CADPrompt.** Table 5 reports results on CADPrompt (Alrashedy et al., 2025), a widely adopted text-to-CAD benchmark (Khan et al., 2024b). SPADA with Gemini-3-flash achieves the best performance across all metrics: 0% IR, 95.4% IoU, and CD of 35.4 (44.8), outperforming specialized methods like MEDA (94.1% IoU, 55.5 CD) and CADCodeVerify (94.4% IoU, 127.0 CD). This contrast validates that our benchmark's lower absolute scores stem from task difficulty rather than model deficiency. The performance gap underscores the fundamental challenge posed by SPADA-Bench-Verified, where coupled parametric constraints and multi-part spatial reasoning push current frontier models to their limits.

## 5.3. Ablation Studies

**Feedback component contribution.** Agent runs are time-consuming and costly; thus, we conduct ablations on a smaller subset of 500 assembly tasks to isolate the contribution of each feedback component (Table 6). Moving from LLM-only reasoning to editable files and terminal feedback raises APR from 11.2% to 16.8%, mainly by reducing in-

*Table 5.* **Results on CADPrompt Benchmark.** CD and HDD are scaled by $10^{-3}$. Best in **bold**, second-best underlined. For CADCodeVerify, we compare against their best reported results.

| Method | IR (%)↓ | IoU (%)↑ | CD↓ | HDD↓ |
|---|---|---|---|---|
| *Zero-shot LLMs & VLMs* | | | | |
| Qwen3-VL-235B | 8.5 | 82.4 | 171.6 (155.4) | 521.3 (485) |
| Gemini-3-flash | 1.0 | 88.2 | 150.7 (137.2) | 446 (412) |
| *Text-to-CAD Methods* | | | | |
| CAD-Coder | 14.5 | 68.4 | 328.3 (287.2) | 893.3 (785.5) |
| CADCodeVerify | 3.5 | 94.4 | 127.0 (135.0) | 419.2 (356.4) |
| MEDA | 1.0 | 94.1 | 55.5 (95.0) | 262.8 (401.0) |
| SPADA (Ours) | | | | |
| w/ Qwen3-235B | 1.0 | 89.4 | 147.1 (135.3) | 423.3 (348.3) |
| w/ Gemini-3-flash | **0.0** | **95.4** | **35.4 (44.8)** | **202.3 (314.2)** |

*Table 6.* **Ablation on feedback components (500 assembly tasks).** CD is scaled by $10^{-3}$. Best in **bold**, second-best underlined.

| Configuration | IR (%)↓ | CD↓ | PR (%)↑ | APR (%)↑ |
|---|---|---|---|---|
| LLM-only (no tools) | 9.8 | 38.7 (32.4) | 20.7 | 11.2 |
| + general tools | 6.5 | 24.3 (20.8) | 28.3 | 16.8 |
| + spec | 4.8 | 14.6 (12.3) | 35.9 | 24.1 |
| + inspect | 3.7 | 8.9 (7.5) | 42.5 | 31.5 |
| **SPADA (Full)** | **2.7** | **2.8 (2.3)** | **51.8** | **39.7** |

*Table 7.* **Self-refinement comparison (100 parts / 100 assemblies).** Best in **bold**, second-best underlined.

| Method | IR (%)↓ | PR (%)↑ | APR (%)↑ |
|---|---|---|---|
| CAD-Coder | 13.5 | 17.8 | 9.5 |
| CAD-Coder w/ Reflexion | 10.0 | 22.5 | 13.0 |
| Gemini-3-flash w/ Reflexion | 5.0 | 44.2 | 27.5 |
| SPADA w/o `verifier` | 2.5 | 54.2 | 45.0 |
| **SPADA** | **1.5** | **66.0** | **53.0** |

valid code and enabling basic repair. The `spec` tool adds 7.3 APR by converting vague intent into explicit parameter and relation predicates. The `inspect` tool adds visual and metadata feedback, improving CD from 14.6 to 8.9 and APR to 31.5%. The `verifier` tool gives the largest single addition, adding 8.2 APR and 9.3 PR by turning residual geometric and relational errors into localized, quantitative repair targets.

**SPADA Against Generic Self-refinement.** To isolate SPADA from generic iterative refinement, we also compare against Reflexion-style self-refinement (Shinn et al., 2023) on a 100-part and 100-assembly subset (Table 7). Generic self-refinement improves CAD-Coder by 3.5 APR and gives Gemini-3-flash a stronger 27.5 APR, showing that iterative repair is useful by itself. However, SPADA without the verifier already exceeds Gemini-3-flash w/ Reflexion by 17.5 APR, indicating that structured specification and CAD-aware inspection provide more actionable state than free-form verbal reflection. Adding the verifier contributes another 8.0 APR, confirming that deterministic test feedback is complementary to the agent scaffold.

### 5.4. Discussion

**Benchmark Challenge Analysis.** The performance gap between CADPrompt (Table 5) and SPADA-Bench-Verified (Tables 3 and 4) reveals fundamental differences in task complexity. SPADA-Bench-Verified's coupled parametric con-

straints, multi-part interactions, and stricter verification requirements expose fundamental limitations in current models' spatial reasoning and constraint satisfaction capabilities. Even with verifier-guided refinement, SPADA achieves only 52.3% APR on parts and 41.9% on assemblies, indicating substantial room for improvement in parametric CAD generation.

**Limitations.** Our method still struggles with certain challenging cases. First, outdated API errors arise due to inherent knowledge limitations—some deprecated APIs cannot be fixed by non-compatible models even when the verifier observes the errors. Second, hard cases involving complex assemblies remain challenging, especially those involving everyday objects not well covered by libraries. Complex structures composed of multiple sub-components sometimes lead to visible spatial misalignments between modules. Third, SPADA-Bench-Verified covers code-based CAD methods, while proprietary CAD formats and their adaptation to code-based CAD remain unexplored. Appendix A.6 provides representative failure modes and iteration-limit cases.

## 6. Conclusion

We presented SPADA, a test-driven agentic framework for parametric CAD assembly generation that bridges natural language specifications and executable engineering designs through sandboxed execution and extensible verifiers. Our experiments demonstrate that verifier-guided iteration substantially improves assembly constraint satisfaction compared to zero-shot approaches. Alongside the agent, we release SPADA-Bench-Verified, a challenging benchmark featuring multi-part assemblies with coupled parametric constraints that expose current frontier model limitations in spatial reasoning. Despite improvements, absolute performance remains modest, underscoring the fundamental difficulty of parametric assembly generation and motivating future work on hierarchical planning, efficient verification, and tighter integration of spatial reasoning capabilities.

## Acknowledgement

This work was supported by the National Natural Science Foundation of China under Grant Nos. 62573141 and W2533151; the Science and Technology Planning Project of Guangdong Province of China under Grant No. 2024A0505040024; and the Science and Technology Program of Guangzhou, China under Grant No. 2024A04J6301.

## Impact Statement

This work advances language-grounded CAD generation for editable mechanical designs and assemblies. The benchmark and sandboxed execution environment support reproducible evaluation of geometric validity, constraint satisfaction, and code quality. Broader access to CAD automation could reduce entry barriers for prototyping, but it also raises risks around unsafe mechanical designs, misuse of generated parts, and overreliance on automated verification. We mitigate these risks by emphasizing deterministic checks, sandboxed execution, and explicit limitations; generated designs should still undergo human engineering review before manufacturing or deployment.

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

# "SPADA: A Verifiable Test-Driven Agent for Controllable Parametric CAD Assembly Generation"

## Overview of Supplementary Material

This appendix provides: **Appendix A** details on SPADA agent design, prompts, and tool interfaces; **Appendix B** execution environment architecture and sandbox implementation; and **Appendix C** benchmark construction, task structure, metric definitions, verifier audits, additional analyses, and qualitative results.

## A. SPADA: Self-testing Parametric Assembly Design Agent Details

### A.1. System Prompt

The system prompt defines the agent's role, output format, workflow, and available tools.

---

**System Prompt**

```
You are SPADA, an expert CAD coding assistant that interacts with a computer to solve tasks in OpenSCAD,
CadQuery, or build123d.
<ROLE>
- Produce clean, parametric, manufacturable CAD models in the requested language.
- Preserve the user's intent; do not add unrelated constraints.
- Prefer millimeter units, named parameters, and reusable modules.
<DELIVERABLE>
- main.scad is the entry point for OpenSCAD tasks; main.py is the entry point for CadQuery/build123d
  tasks.
- Return only the code for the requested entry point.
<WORKFLOW>
1. Extract requirements (and spec if needed).
2. Write the entry point (main.scad or main.py).
3. Use inspect to validate geometry.
4. Use verifier to write and run tests (unless explicitly told to skip tests).
<TOOLING>
- terminal: explore files and run commands.
- file_editor: create or update files in the workspace.
- spec: extract a structured model spec from a prompt.
- inspect: compile the current model and report geometry stats.
- verifier: write and run pytest/SpecSCAD tests.
```

---

### A.2. Tool Interface and Implementation

**Core tools.** `terminal` runs shell commands in the active workspace. `file_editor` writes and patches files in the workspace. If a core tool is not provided by the execution host, SPADA registers a fallback implementation to ensure consistent behavior across deployment environments.

**Spec extraction (`spec`).** The `spec` tool extracts a structured `ModelSpec` from the natural-language instruction. The specification includes: a task name and intent description, global and part-level parameters with types and defaults, a list of verification checks (compilation, bounding-box limits, manifold requirements), pytest-ready test assertions, and soft constraints that serve as a progress tracker.

```
Spec Extraction Prompt

You are a requirements engineer for CAD tasks. Your job is to turn a natural-language request into
a *verifiable* model specification for a spada-task that can be checked with pytest-based tests and
simple code/structure checks.
<RULES>
- Output must be STRICT JSON (no markdown, no backticks, no commentary).
- Make the spec verifiable with tools available after generation.
- Only include checks that are realistically testable.
- Prefer millimeters for units unless the instruction specifies otherwise.
- For any parameter that is relevant but not specified by the user, choose a feasible numeric default
(do not leave it null or "TBD").
<OUTPUT_SCHEMA>
{
  "name": "short_name",
  "intent": "1-3 sentence assembly description",
  "parameters": {"param": {"type": "float", "default": 20.0}},
  "parts": {"part_name": {"desc": "...", "parameters": {...}}},
  "assembly_scheme": {"components": [...], "constraints": [...]},
  "verifications": [...],
  "tests": [...], "soft_constraints": [...]
}
```

**Geometry inspection (`inspect`).** The inspect tool compiles the entry point and returns structured geometry metadata: bounding box dimensions, volume, manifold status, triangle count, and compile time. When rendering is enabled, it also returns base64-encoded images from specified viewpoints. This feedback helps the agent catch gross errors (wrong scale, non-manifold geometry) before running full constraint verification.

**Test verification (`verifier`).** The verifier tool writes test modules into a tests/ directory and executes them. For OpenSCAD tasks, it generates SpecSCAD-format tests; for Python CAD tasks, it generates pytest modules and runs pytest. The tool returns structured logs and pass/fail status for each test. If test generation fails, the agent injects a minimal fallback test to ensure deterministic behavior.

### A.3. Model Configurations

All models are accessed via their official APIs with consistent settings to ensure fair comparison. For all experiments, we set temperature=0 to ensure deterministic outputs and reproducibility. Maximum output tokens are set to 8,192 to accommodate long CAD code generation. For thinking model variants (*e.g.*, o1-style), we apply equivalent thinking budgets across models. To prevent endless refinement loops, we impose a 5-minute (300 second) wall-clock timeout per task. The agent performs multiple compile-inspect-verify cycles within this budget when time remains. On average, SPADA uses 4.1 iterations per task with the full tool suite.

### A.4. Cost Analysis

Cost varies with model pricing and the number of repair iterations. In our runs, SPADA uses about four iterations per task on average. The approximate per-task API cost is 0.03–0.15 USD for Gemini-3-flash, 0.10–0.50 USD for GPT-5, and 0.02–0.08 USD for Qwen3-VL-235B. These ranges reflect variation in prompt length, generated code length, visual feedback, and iteration count.

### A.5. Scalability and Generalization

The verifier generation pipeline scales because tests are derived from structured task metadata, golden workspaces, and executable geometric predicates. SPADA-Bench-Verified uses this pipeline to construct 11.7k tasks across OpenSCAD, CadQuery, and Build123d. Extending the benchmark to new code-based CAD ecosystems requires exposing equivalent geometry APIs and adding backend-specific templates for interface, geometry, and constraint checks.

For SPADA itself, scaling inference is mainly an execution-infrastructure problem: candidates must be compiled, rendered, and tested in isolated sandboxes. The framework is parallel by construction because each task and each candidate can be

executed independently. The current evidence validates programmatic CAD representations; non-code CAD formats and proprietary environments remain outside the validated scope.

### A.6. Failure Analysis

The main text summarizes the limitation categories. Here we expand the iteration-limit cases observed during runs. Outdated API knowledge led some models to repeat deprecated or incompatible CAD calls after error feedback. Complex assemblies with coupled constraints produced partial repairs that fixed one relation while breaking another. Objects poorly covered by standard CAD libraries, such as organic or everyday shapes with many sub-components, produced spatial misalignment between modules. Some models also confused constructive operations, such as extrusion versus cutting, yielding valid but semantically wrong geometry. These failures show that verifier-guided repair improves local correction, while root-cause planning for multi-constraint edits remains unresolved.

## B. Environment Details

This section describes the execution environment that enables SPADA to compile, inspect, and verify CAD code across multiple backends. The environment design prioritizes reproducibility, isolation, and structured feedback.

### B.1. Execution Harness Architecture

The execution harness follows a gym-like step-based interaction paradigm. At each step, the agent submits an action specifying: (i) a mapping of workspace-relative filenames to contents, and (ii) an entry-point file to execute or compile. The harness writes these files into an isolated workspace, invokes the appropriate backend, and returns a structured observation containing success status, logs, and generated artifact names.

**Action space.** All backends implement the same primary action schema:

- `files: dict[str, str]` — a map from workspace-relative paths to file contents.

- `entry_point: str` — the file to execute/compile.

This unified interface allows the agent to operate identically across OpenSCAD, CadQuery, and Build123d backends.

**Observation space.** Each step returns a structured JSON observation with the following fields:

- `success` (bool): whether the backend produced expected artifacts.

- `logs` (str): combined stdout/stderr from compilation/execution.

- `artifacts` (list[str]): generated filenames (STL, PNG, etc.).

- `metadata` (dict): execution metadata including geometry statistics.

### B.2. Unified Code-based CAD Sandbox

For geometry inspection and rendering, SPADA uses a unified CAD sandbox interface to provide language-agnostic execution and feedback. We build on OpenEnv [3], an open-source agentic execution sandbox with clean abstractions that support isolated execution, structured observations, and flexible backend integration.

The agent sends a JSON payload containing the execution mode (`geometry` or `render`), the CAD engine identifier, the entry-point filename, and a complete workspace file map. The sandbox returns geometry metadata and, for rendering requests, base64-encoded images.

---

[3] `https://github.com/meta-pytorch/OpenEnv`

Sandbox Request Payload Structure

```
{
  "mode": "geometry",
  "engine": "openscad",
  "entry_point": "main.scad",
  "files": {
    "main.scad": "// CAD code here..."
  }
}
```

## B.3. Execution Pipeline

Each environment `step()` call proceeds through four guarded stages:

**(1) File synchronization.** The harness validates all paths to prevent directory traversal attacks, then writes agent-provided files into the isolated workspace. Files are written atomically to prevent partial writes from corrupting the workspace state.

**(2) Entry resolution.** The harness resolves the `entry_point` string to an absolute path inside the workspace. If the entry point does not exist, the step fails immediately with a descriptive error.

**(3) Compilation/Execution.** The harness invokes the backend executable with appropriate arguments and captures output. For OpenSCAD, this means running `openscad -o output.stl main.scad`; for Python CAD, this means running the entry point with the appropriate Python environment. Execution is subject to configurable timeouts (default: 60 seconds).

**(4) Observation packing.** The harness constructs and returns a structured JSON observation. On success, this includes artifact paths and geometry metadata; on failure, this includes error context with line numbers and suggested fixes when available.

## B.4. Observation Examples

The following examples illustrate the structured feedback that enables targeted agent repairs.

**Successful compilation.** When compilation succeeds, the observation includes geometry metadata that the agent uses to validate dimensions and structure:

Observation: Successful Compilation

```
{
  "success": true,
  "stdout": "Compiling design...",
  "artifacts": {
    "stl": "/workspace/output.stl",
    "png": "/workspace/render.png"
  },
  "metadata": {
    "bbox": {"min": [0,0,0], "max": [50,50,20]},
    "volume_mm3": 12500.0,
    "is_manifold": true,
    "triangle_count": 2048,
    "compile_time_s": 1.42
  }
}
```

The metadata includes bounding box, volume, manifold status, and triangle count, enabling the agent to validate geometry before running full verification.

**Compilation error.** When compilation fails, the observation includes structured error context that enables targeted repair:

**Observation: Parser Error**

```
{
  "success": false,
  "stderr": "ERROR: Parser error: syntax error in file 'main.scad', line 15",
  "error_context": {
   "line": 15,
   "column": 8,
   "snippet": "cube([10, 20 30]); // missing comma"
  },
  "suggestion": "Check for missing delimiters"
}
```

The error context includes line number, column, code snippet, and a suggested fix, enabling the agent to make precise repairs without rewriting the entire file.

The structured error context—including line numbers, code snippets, and suggestions—enables the agent to make minimal targeted edits rather than regenerating the entire file. This approach preserves previously correct code and improves convergence.

## C. SPADA-Bench-Verified Details

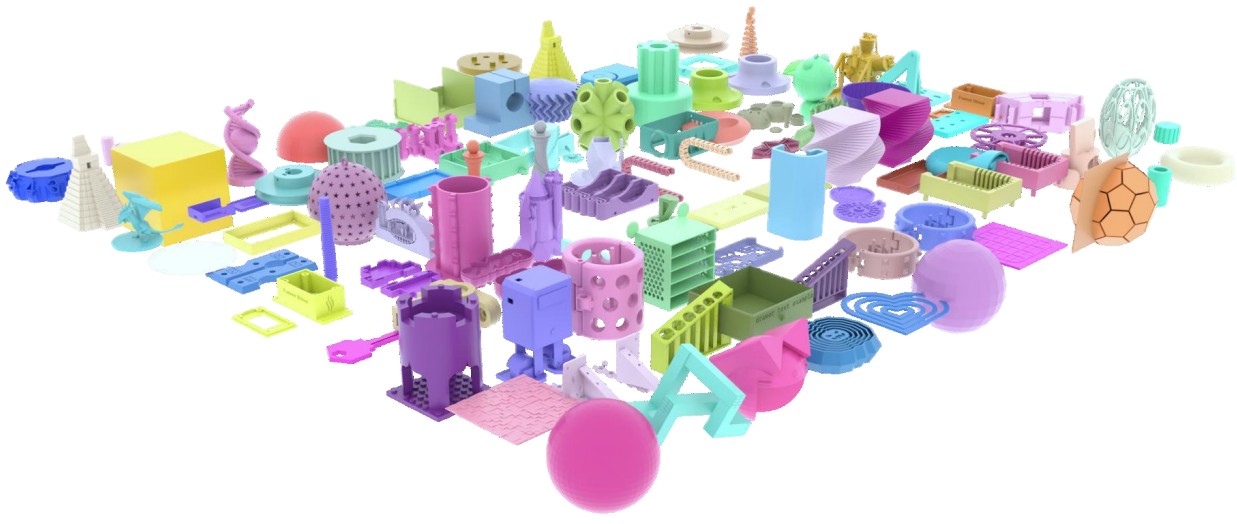

*Figure 5.* **Random-sampled models from SPADA-Bench-Verified.** The benchmark draws from diverse data sources across active CAD communities, covering a wide range of editable parametric models: mechanical components (gears, brackets, enclosures), functional assemblies (hinges, clamps, connectors), and everyday objects (containers, stands, organizers). Each model features explicit parameters, modular structure, and engineering-grade constraints.

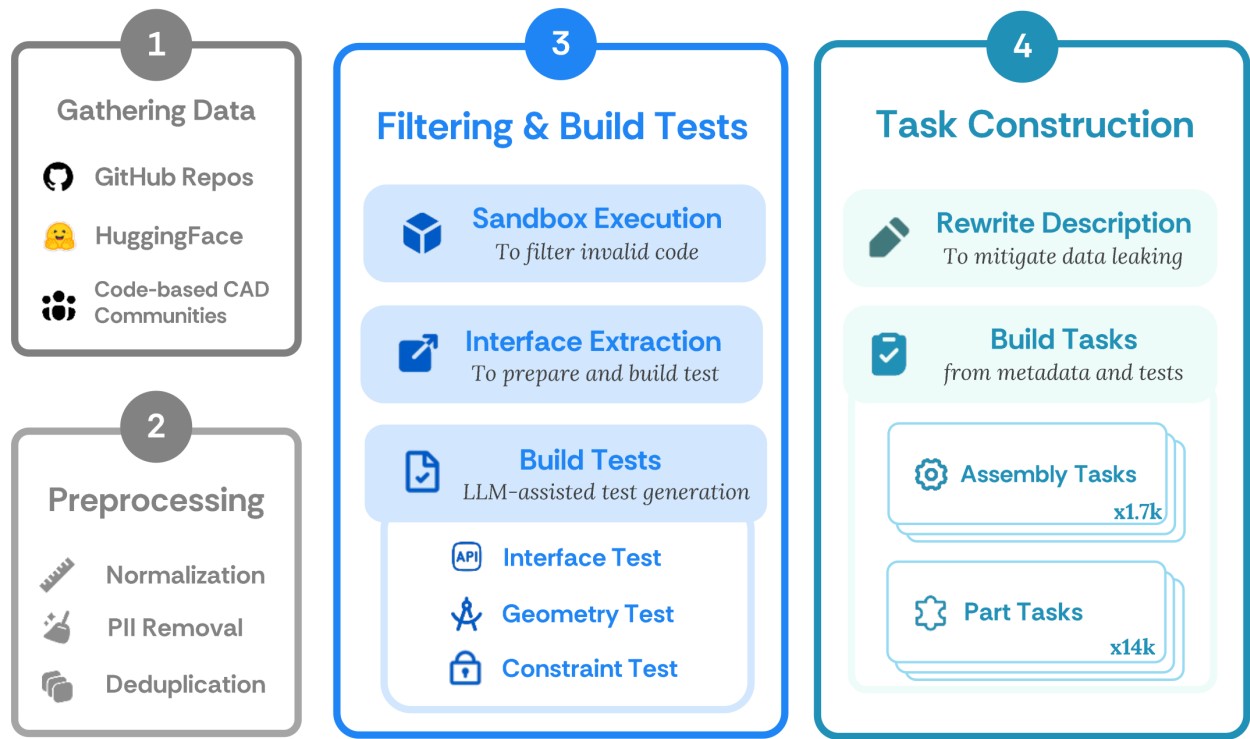

*Figure 6.* **SPADA-Bench-Verified Construction Pipeline.** Three-stage pipeline from raw corpus to verified benchmark: execution-based filtering ensures validity, LLM-assisted rewriting mitigates data leakage, and automated test generation produces deterministic verifiers. The final dataset (1.7k assemblies, 10k parts) enables objective pass/fail evaluation through interface, geometry, and constraint tests.

### C.1. Task Structure

We use harbor [4] as our evaluation scaffold. In SPADA-Bench-Verified, each benchmark task is packaged as a self-contained unit with a standardized directory structure:

```
Task Directory Structure

<task_id>/
  instruction.md                                          # Natural language prompt
  config.toml                                                  # Task configuration
  environment/                                              # Execution environment
    DOCKERFILE
    requirements.toml
  tests/
    verify.py                                                     # Verification logic
    golden.scad                                          # Reference implementation
  solution/
    /parts                                                      # Optional part files
      part1.scad
      part2.scad
    main.scad                                                            # Entry point
```

**Single-part Task.** In single-part tasks, the `instruction.md` file contains a natural-language description of the desired part, including parameters and constraints.

---

[4]`https://harborframework.com/docs`

---

**Single-part Task Specification Schema**

```
- name: string, short name for the model
- intent: string, 1-3 sentences describing the intended assembly
- parameters: object {key: {type, default}}
- parts: object where each key is a part name:
  - desc: string, brief description
  - parameters: object {key: {type, default}}
- assembly_scheme: object with:
  - components: list of {part_name, instance_name, overrides}
  - constraints: list of {type, details}
- verifications: list of checks (code_contains, compiles, bbox_size, manifold)
```

---

**Assembly Task.** Each assembly task in SPADA-Bench-Verified is defined using a standardized schema that captures the essential elements of parametric CAD design. An automatic converter transforms this schema into the task structure above.

---

**Assembly Schema Structure**

```
name: string                                        # short model name
intent: string                          # 1-3 sentences describing intent

units:
 length: {type: string, default: "mm"}
 angle: {type: string, default: "deg"}

parameters: object                                  # global parameters
 <parameter_name>:
   type: {float, int, string, etc.}
   default: value

parts: object                                       # part definitions
 <part_name>:
   desc: string
   parameters: object
   constraints: list

assembly_scheme:
 components: list                                    # part instances
   - part_name: string
     instance_name: string
     parameter_overrides: object
 links: list                              # relations between components
   - type: {mate, align, fix}
     details: object

sources: string                              # original source attribution
```

---

## C.2. Preprocessing Details

**Data Sources.** We collect CAD programs from code-centric CAD communities, GitHub repositories, and Hugging Face Datasets across the three most widely adopted code-based CAD backends: OpenSCAD, CadQuery, and Build123d. We retain only samples with compatible redistribution licenses, and we store license metadata for each task. Specifically, the sources include:

- **Communities**: We scrape user-contributed CAD programs from five active CAD communities: Thingiverse[5], Cults3D[6],

---

[5]https://www.thingiverse.com/
[6]https://cults3d.com/

CGTrader[7], MakerWorld[8], and Thangs[9];

- **GitHub / Huggingface Repos**: We mainly use Stack-V2-dedup as a starting point. We filter repositories that contain CAD programs in OpenSCAD, CadQuery, and Build123d by searching for characteristic file extensions and import statements. We also manually curate from awesome-series GitHub repositories that focus on code-based CAD designs;

- **Manual Selection**: We manually collect CAD programs from educational resources, tutorials, and example galleries to ensure coverage of fundamental design patterns.

**Preprocessing and Candidate Pool.** Raw programs are normalized into a consistent workspace layout per CAD backend. We remove personally identifying strings (for example, emails and local user paths) using pattern matching. We remove near-duplicates using LSH-based matching over token shingles. We then execute each program in a unified sandbox with pinned dependencies, backend versions, and execution flags; only programs that run successfully and produce valid geometry artifacts are retained. The resulting executable pool contains 200K programs.

**Interface Contract Extraction.** For each part task, we extract an interface contract $\Gamma$ that includes: (i) exported symbol name (module/function/class), (ii) parameter names and defaults, (iii) parameter types/units when available from the source language or annotations.

**Rewrite Description and Interface.** In early experiments, we found data contamination issues when using original interfaces and related descriptions directly. Following the approach of GSM-Symbolic (Mirzadeh et al., 2024), we use a rewriting LLM, gpt-oss-120b, to paraphrase the original task descriptions while preserving the core intent.

**Difficulty Control.** Inspired by AutoCodebench (Chou et al., 2025), we use a moderately capable code model to filter overly easy tasks. Specifically, we use `gpt-oss-20b` for its high inference speed and coding ability. We let `gpt-oss-20b` generate code for all candidate tasks with a single attempt. If the generated code passes all interface tests, we mark the task as easy and filter it out.

### C.3. Test Categories

The `verifier` performs semantic verification over executed CAD outputs rather than source-level syntax checking. A syntax checker accepts a program once it parses and compiles; the verifier executes the workspace and checks whether the resulting interface, geometry, and assembly state satisfy constraints that can be written as executable assertions. Constraints that describe non-executable aesthetic intent are kept out of pass/fail scoring.

We organize these assertions into three test categories:

1. **Interface tests** ($\mathcal{T}_{\text{int}}$): Verify that required APIs exist, parameter names and defaults match the contract, and the entry point instantiates with the provided configurations.

2. **Geometry tests** ($\mathcal{T}_{\text{geo}}$): Verify validity, coarse dimensions, metric agreement with the reference geometry, and shape-defining parameter relationships. For example, a thread task checks that crest position, pitch diameter, crest diameter, thread height, pitch, and flank angle remain mutually consistent within tolerance. We use $CD < 1 \times 10^{-3}$ as the geometric agreement threshold (Guan et al., 2025).

3. **Constraint tests** ($\mathcal{T}_{\text{cons}}$): Verify assembly predicates over the realized assembly state $G$, including concentricity, coplanarity, parallelism, perpendicularity, mate/align relations, distance and angle bounds, clearance bounds, and collision-free placement.

Failures are localized to the violated assertion and reported with measured values when available. For example, a program that compiles cleanly still fails if it renames a required parameter, produces degenerate geometry, violates a thread-parameter relationship, reports a pitch diameter of 22mm when 24mm is expected, or leaves a 0.3mm collision between parts. These quantitative signals explain why verifier feedback gives the largest single gain in the ablation in Table 6.

---

[7]https://www.cgtrader.com/
[8]https://makerworld.com/
[9]https://thangs.com/

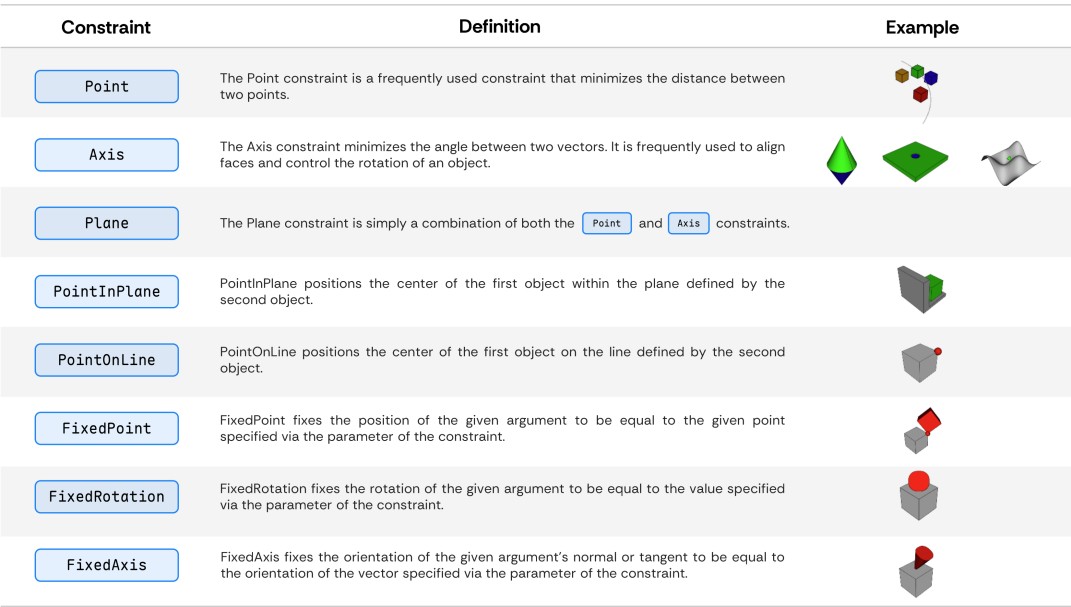

*Figure 7.* **Assembly Constraint Examples.** In practice, these constraints combine to form complex assembly relations.

Verifier generation combines deterministic templates with backend-specific test mechanisms. For OpenSCAD, we use community-compatible assertion patterns from its built-in testing workflow [10]. For assembly constraints, structured templates compose primitive predicates into complex relations, and each failed constraint reports its id, measured value, expected bound, and involved instances.

We also audited verifier quality on 200 sampled tasks, consisting of 100 part tasks and 100 assembly tasks. Interface and geometry tests are generated by deterministic scripts from the golden reference signature and geometry templates, and the templates were reviewed by a CAD expert before deployment. Constraint tests are generated from structured templates with LLM assistance and then checked by human review. The audit found 98.5% accuracy across sampled tests. The observed errors were overly strict tolerances relative to part scale and missing edge-case constraints for complex inter-part relations; all identified errors were corrected in the benchmark.

### C.4. LLM-as-a-Judge Evaluation

For large-scale evaluation of dataset quality, following previous work (Govindarajan et al., 2026), we use LLM-as-a-judge to assess human-likeness of CAD code. We use `gpt-oss-20b` as the judge, which outputs integer scores from 0 to 5, where 5 indicates code that is most human-like.

---

**LLM-as-a-Judge System Prompt**

```
You are a strict rater for whether a code snippet looks human-written.
Task: Given a CAD script, output a human-likeness score as an integer from 0 to 5, where 5 is most like
human.
Score meanings:
0: Almost surely an automatic primitive dump; no meaningful structure.
1: Mostly automatic; minor human touches but still dump-like.
```

---

[10]https://github.com/openscad/openscad/issues/4075

```
2: Mixed; some structure but many dump signals.
3: Plausible human; structure exists but looks somewhat mechanical.
4: Strongly human-like; modular and readable with minor issues.
5: Strongly human-like; good abstraction, parameters, maintainable style.
What to look for (human-like signals):
- Parameterization: named variables for key dimensions.
- Decomposition: modules/functions for repeated parts.
- Control structure: loops for symmetry and repetition.
- Naming: meaningful identifiers; consistent terminology.
- Readability: consistent indentation; spacing; small blocks.
- Comments: helpful and not excessive.
- Maintainability: avoids long repeated blocks; uses helper modules.
Automatic / dump-like signals:
- Many repeated translate/rotate/cube lines with small numeric changes.
- "BLOCK N" markers or other auto-export comments.
- Many float literals with fixed formatting.
- Little or no variables, loops, modules, or reuse.
- Very high repetition of near-identical lines.
Output format requirements:
- Output ONLY valid JSON.
- No extra text outside JSON.
- JSON keys: score (int 0..5), rationale (string), confidence (float 0..1), signals (object).
```

### LLM-as-a-Judge User Prompt Template

```
Code to rate:
--BEGIN CODE--
{code}
--END CODE--
Return ONLY the JSON object with keys: score, rationale, confidence, signals.
In signals, include short, concrete evidence such as: modules, assignments, loops, repetition patterns,
numeric-literal patterns, comments, formatting consistency, and any dump markers.
```

## C.5. Geometric Fidelity Metric Details

Both generated and ground truth CAD models are first exported to STL format, then converted to point clouds by uniformly sampling **2000 points** from mesh surfaces. We apply **ICP (Iterative Closest Point) alignment** (Besl & McKay, 1992) for optimal rigid transformation before computing distances.

**Chamfer Distance (CD)**    measures bidirectional average nearest-neighbor distance between point clouds $P$ (generated) and $Q$ (ground truth):

$$\text{CD}(P,Q) = \frac{1}{2|P|} \sum_{p \in P} \min_{q \in Q} \|p - q\|_2 + \frac{1}{2|Q|} \sum_{q \in Q} \min_{p \in P} \|q - p\|_2 \tag{5}$$

**Hausdorff Distance (HDD)**    measures worst-case geometric deviation:

$$\text{HDD}(P,Q) = \max \left\{ \sup_{p \in P} \inf_{q \in Q} \|p - q\|_2, \sup_{q \in Q} \inf_{p \in P} \|q - p\|_2 \right\} \tag{6}$$

## C.6. Fine-grained Benchmark Breakdown

Tables 8 and 9 report performance by CAD language and assembly constraint count. OpenSCAD is easiest, likely because public OpenSCAD code is more common in model pretraining corpora. CadQuery and Build123d have lower APR, and performance degrades as the number of assembly constraints increases.

*Table 8.* **Per-language performance of SPADA with Gemini-3-flash.**

| Task | CAD Language | #Tasks | IR (%)↓ | PR (%)↑ | APR (%)↑ |
|------|-------------|--------|---------|---------|----------|
| Part | OpenSCAD | 6,824 | 1.2 | 69.3 | 56.4 |
| Part | CadQuery | 2,107 | 2.7 | 58.7 | 44.8 |
| Part | Build123d | 1,069 | 3.9 | 56.5 | 40.8 |
| Assembly | OpenSCAD | 1,183 | 1.8 | 58.3 | 46.2 |
| Assembly | CadQuery | 347 | 3.6 | 46.5 | 33.5 |
| Assembly | Build123d | 170 | 4.1 | 41.4 | 29.1 |

*Table 9.* **Assembly performance by number of constraints.**

| #Constraints | #Tasks | PR (%)↑ | APR (%)↑ |
|-------------|--------|---------|----------|
| 1–3 | 620 | 63.8 | 52.4 |
| 4–6 | 710 | 53.0 | 40.1 |
| 7+ | 370 | 40.3 | 27.6 |

## C.7. Expert Preference Study

We conducted a blinded pairwise preference study following the LMArena protocol style (Chiang et al., 2024). We sampled 100 assembly tasks and recruited five experts familiar with code-based CAD and mechanical engineering. For each task, experts saw anonymized, randomly ordered generated workspaces and selected the preferred result based on constraint satisfaction, structural correctness, and parametric editability. The study produced 796 pairwise judgments. We aggregate preferences with a Bradley–Terry model and compute confidence intervals with bootstrap resampling.

*Table 10.* **Expert pairwise preference study on assembly tasks.** Higher Arena Score is better.

| Rank | Method | Arena Score | Votes |
|------|--------|-------------|-------|
| 1 | **SPADA (Ours)** | **1147.5 ± 22.4** | 202 |
| 2 | CADCodeVerify | 1089.8 ± 24.7 | 200 |
| 3 | Gemini-3-flash | 1021.3 ± 26.1 | 198 |
| 4 | MEDA | 943.2 ± 28.3 | 196 |

## C.8. Training and Reward-based Adaptation

SPADA-Bench-Verified is compatible with training because each task provides structured intent and workspace pairs. We therefore evaluate whether the verifier can support both supervised fine-tuning (SFT) and reward-based adaptation. We fine-tune Qwen3-VL-30B-A3B with 16-bit LoRA and evaluate on a separate 200-task split. We also run rejection sampling fine-tuning (RFT), where candidates that pass all golden tests are retained, and DPO (Rafailov et al., 2023), where verifier pass/fail outcomes define preference pairs. RFT improves over plain SFT across all metrics, confirming that verifier-filtered data provides discriminative supervision. DPO yields a further gain by using preference pairs from verifier outcomes. The gain on top of the SPADA loop is smaller than the one-pass gain, which is expected because training-time verifier supervision and test-time repair address overlapping error modes.

*Table 11.* **Training and verifier-reward adaptation on a 200-task split.** SFT improves one-pass generation, while RFT and DPO show that verifier outcomes provide useful training signals. Best in **bold**, second-best underlined.

| Method | IR (%)↓ | PR (%)↑ | APR (%)↑ |
|--------|---------|---------|----------|
| Zero-shot | 35.0 | 9.4 | 3.5 |
| SFT | 16.5 | 14.9 | 6.5 |
| SFT + RFT | 14.0 | 18.7 | 9.5 |
| SFT + DPO | 13.0 | 20.2 | 11.0 |
| SPADA (Qwen w/ SFT) | 11.5 | 34.8 | 20.5 |
| SPADA (Qwen w/ RFT) | 10.0 | 36.2 | 22.0 |
| SPADA (Qwen w/ DPO) | **9.5** | **37.1** | **23.0** |

## C.9. Additional Qualitative Results

Figures 8 to 11 show additional SPADA outputs across mechanical components, storage and organization objects, household objects, decor objects, and educational objects.

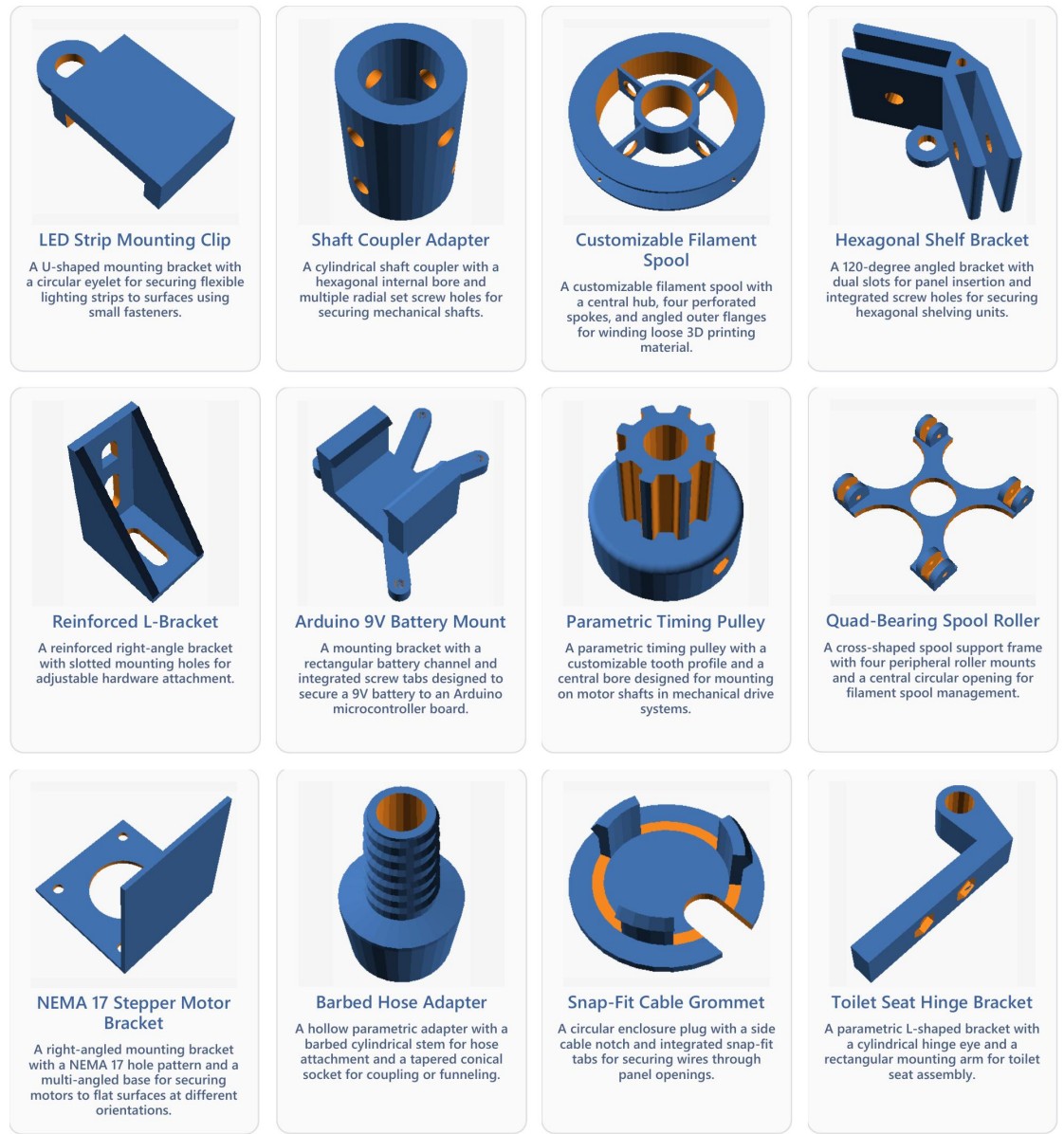

*Figure 8.* Supplementary SPADA generation results: mechanical components.

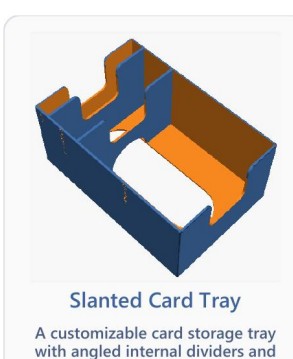

**Slanted Card Tray**

A customizable card storage tray with angled internal dividers and side finger notches for easy deck access.

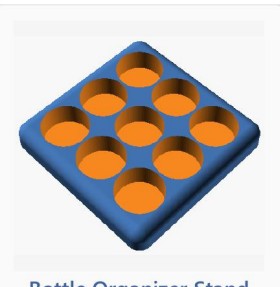

**Bottle Organizer Stand**

A square storage tray with rounded corners featuring nine circular recessed slots arranged in a grid for holding small bottles.

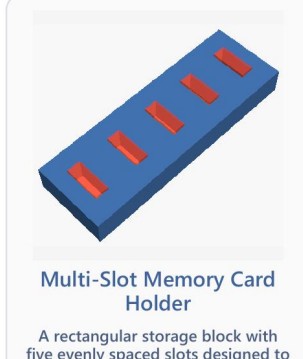

**Board Game Card Holder**

A rectangular storage tray with large side access cutouts and internal support ribs designed for organizing small format tabletop game cards.

**Multi-Slot Memory Card Holder**

A rectangular storage block with five evenly spaced slots designed to hold memory cards or USB drives in an upright position.

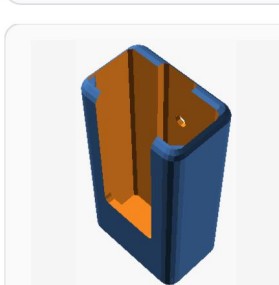

**Wall-Mounted Remote Holster**

A wall-mountable sleeve with a U-shaped profile and three vertical mounting holes designed to securely hold a rectangular remote control.

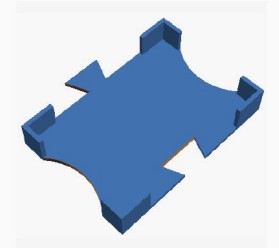

**Interlocking Card Tray**

A modular card tray with corner pillars, semi-circular end cutouts for easy access, and interlocking side joints for connecting multiple holders.

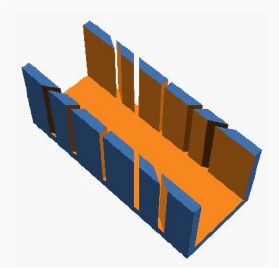

**Multi-Angle Miter Box**

A U-shaped guide channel with vertical and angled slots for manual cutting at ninety, forty-five, and thirty degree angles.

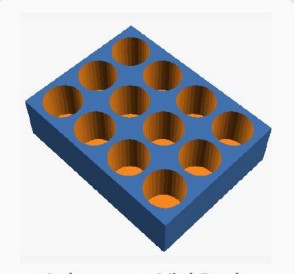

**Laboratory Vial Rack**

A rectangular laboratory rack with a grid of cylindrical wells for holding vials or cuvettes.

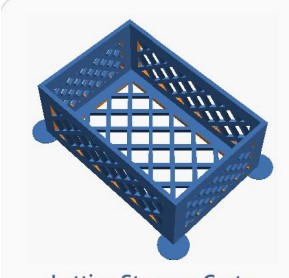

**Lattice Storage Crate**

A rectangular storage crate with diamond lattice walls and circular corner tabs for mounting or stacking.

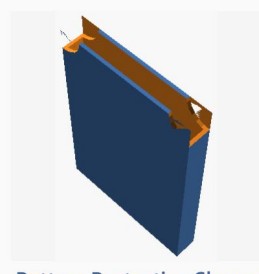

**Battery Protective Sleeve**

A rectangular protective sleeve with corner cutouts designed to securely house and insulate a flat mobile phone battery.

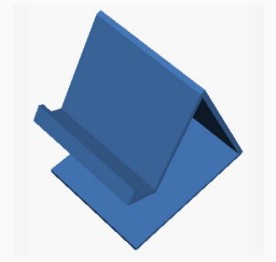

**Angular Phone Stand**

A single-piece device holder with a slanted back support and a front retaining lip.

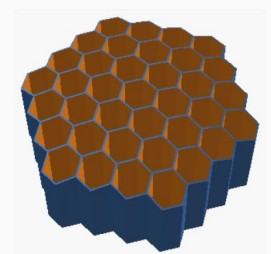

**Honeycomb Pencil Holder**

A hexagonal desk organizer with multiple vertical honeycomb cells for storing pens and pencils.

*Figure 9.* Supplementary SPADA generation results: storage and organization objects.

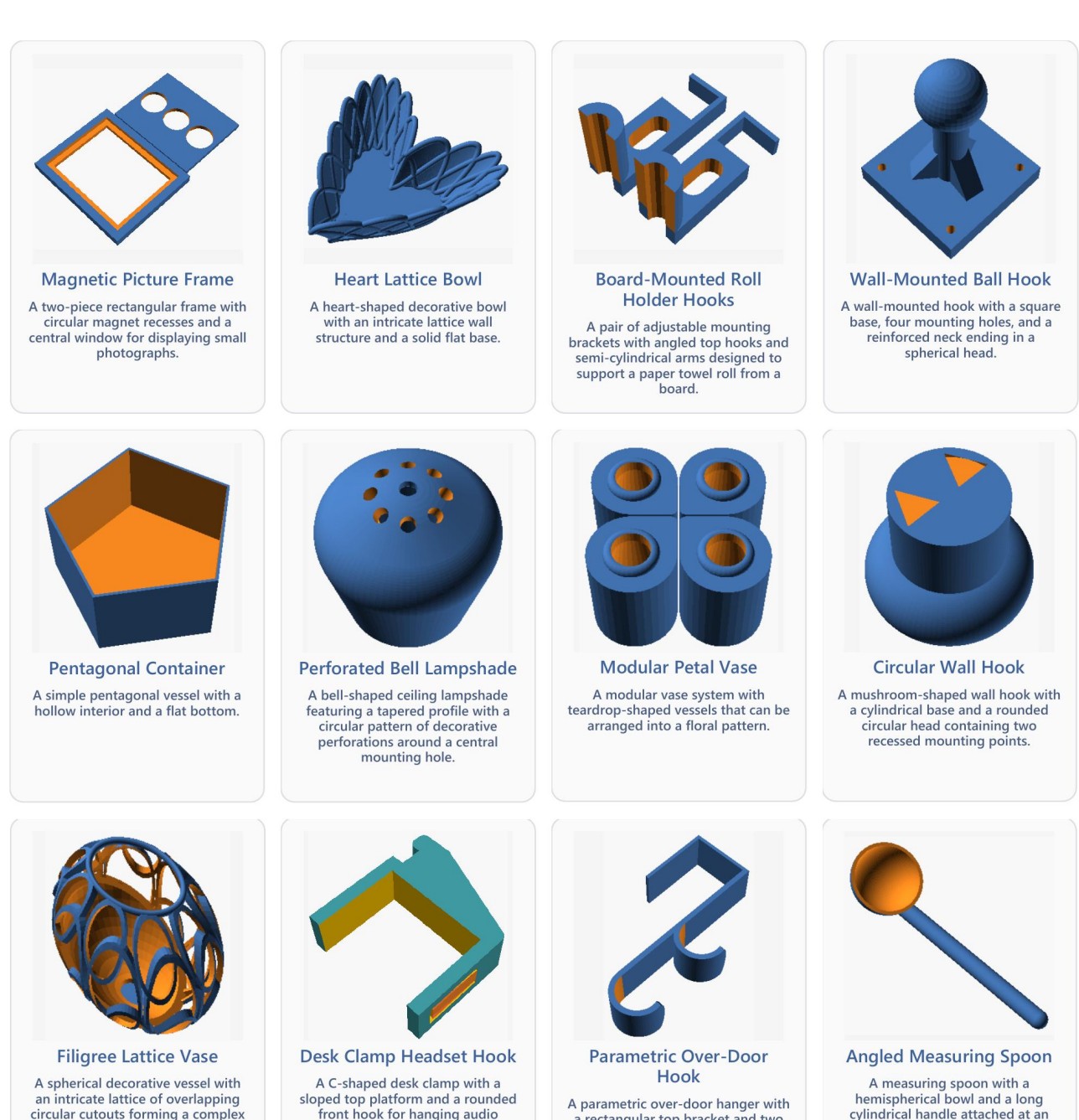

*Figure 10.* Supplementary SPADA generation results: household and decor objects.

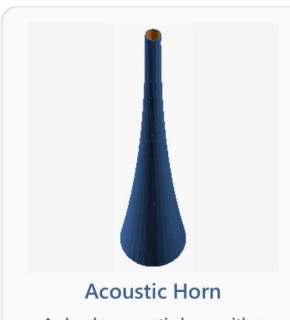

**Acoustic Horn**

A slender acoustic horn with a narrow cylindrical base and a long, gradually widening conical profile.

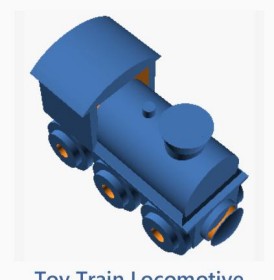

**Toy Train Locomotive**

A stylized toy locomotive body with three axle points, a front smokestack, and integrated mounts for magnetic couplers.

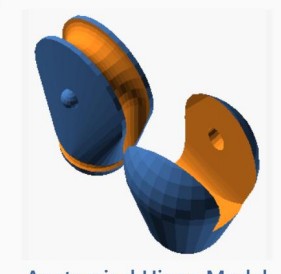

**Anatomical Hinge Model**

A two-part hinged joint with a pivot mechanism and integrated grooves for securing tubes and elastic materials to demonstrate anatomical articulation.

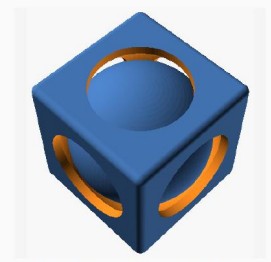

**Captive Sphere Cube**

A hollow rounded cube with circular face openings containing a captive sphere that is larger than the holes.

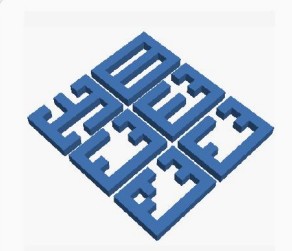

**Gordian Knot Puzzle**

A six-piece interlocking 3D puzzle with intricate internal notches and protrusions designed to slide together into a single compact assembly.

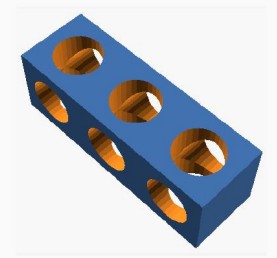

**Perforated Square Beam**

A square structural beam with evenly spaced circular holes on perpendicular faces designed for modular mechanical assembly and robotic skeletons.

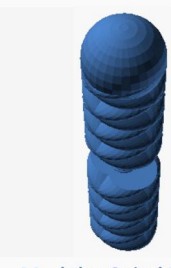

**Modular Spiral Wand**

A modular decorative wand with twisted spiral segments and integrated screw connectors for customizable assembly.

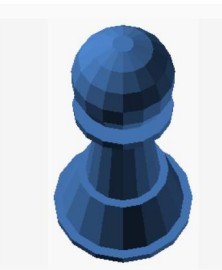

**Low-Poly Chess Pawn**

A low-poly chess pawn with a spherical head, a tapered neck, and a flared base.

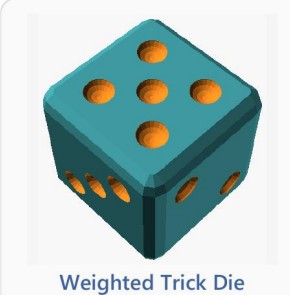

**Weighted Trick Die**

A six-sided die with rounded corners and conical pips featuring an internal hollow chamber to influence the probability of landing on a specific face.

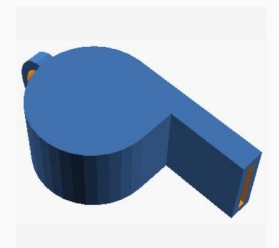

**Customizable Whistle**

A customizable whistle with a cylindrical resonance chamber, a rectangular mouthpiece, and a small attachment loop at the rear.

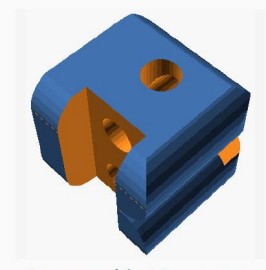

**Anamorphic Monogram Cube**

A solid cube with intersecting letter-shaped extrusions that display different characters when viewed from three perpendicular directions.

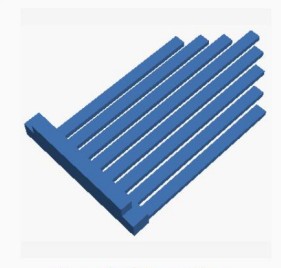

**Thumb Piano Tines**

A musical instrument component with a series of parallel rectangular tines of varying lengths mounted to a transverse base.

*Figure 11.* Supplementary SPADA generation results: educational and play objects.

