# Supplementary Material for Paper
# "SPADA: A Verifiable Test-Driven Agent for Controllable Parametric CAD Assembly Generation"

## Overview of Supplementary Material

This appendix provides: **Appendix A** details on SPADA agent design, prompts, and tool interfaces; **Appendix B** execution environment architecture and sandbox implementation; **Appendix C** benchmark construction, task structure, and metric definitions.

## A. SPADA: Self-testing Parametric Assembly Design Agent Details

### A.1. Design Thinkings and Principle

**Why Code-based CAD?**  Among these representations, *code-based* or programmatic CAD stands out for generative applications. Representing geometry as executable programs—in languages such as OpenSCAD, CadQuery, or Build123d—exposes design intent through standard programming constructs: loops create patterns, functions encapsulate reusable components, and variables define editable dimensions. This programmatic structure provides "white-box" access to the design, enabling automated testing, modular reuse, and direct parameter manipulation. Furthermore, code-based CAD leverages the vast pre-training corpora of Large Language Models, which have seen billions of lines of source code and can generalize programming patterns to geometric domains. Unlike opaque feature trees in graphical CAD systems, code is inspectable, diffable, and version-controllable—properties essential for iterative design and collaborative engineering.

**Code-based Parametric CAD Ecosystem.**  The code-based CAD ecosystem centers on three major languages and their communities. **OpenSCAD** is a declarative, functional language where geometry is defined by combining primitives (cubes, cylinders, spheres) through CSG operations (union, difference, intersection); its active community on Thingiverse and GitHub has produced tens of thousands of parametric designs. **CadQuery** is a Python library that wraps the OpenCASCADE kernel, offering imperative, object-oriented modeling with direct access to B-rep topology; its integration with standard Python tooling makes it popular for engineering automation. **Build123d** is a newer Python framework that provides both imperative and declarative APIs with improved ergonomics over CadQuery. All three support parametric design through explicit variables and functions, produce standard output formats (STL, STEP), and can be executed headlessly in CI/CD pipelines—properties that make them ideal substrates for agentic CAD generation.

**Minimal Tool Design.**  Agent frameworks often proliferate tools to handle diverse scenarios, but tool overload increases ambiguity in tool selection and complicates prompting. SPADA adopts a minimal design: beyond basic file editing and terminal access, only three CAD-specific tools (spec, inspect, verifier) compose the entire workflow. This constraint forces the agent to rely on general reasoning rather than specialized shortcuts, improves reproducibility across backbone models, and simplifies the action space for potential reinforcement learning fine-tuning. Each tool has a single, well-defined responsibility—specification extraction, geometry observation, and test execution—avoiding overlapping functionality.

**Why Test-Driven Development?**  Test-driven development (TDD) provides a natural framework for controllable generation. By synthesizing deterministic tests alongside code, SPADA establishes an executable contract that specifies exactly what "correct" means for each task. This framing offers two key benefits. First, binary pass/fail outcomes eliminate the ambiguity of similarity metrics or VLM judgments, providing consistent evaluation across runs. Second, deterministic rewards are directly compatible with reinforcement learning from verifiable rewards (RLVR): the pass/fail signal can serve as a sparse but unambiguous reward for policy optimization, opening paths to self-improvement beyond prompting alone. Recent work on math reasoning has shown that models can bootstrap from verifiable outcomes; code-based CAD with executable tests offers an analogous structure for geometric domains.

**Verifier Design.**    An deterministic verifier design yields key advantages evident in our results. First, *deterministic outcomes* eliminate VLM interpretation variance, as binary pass/fail provides consistent evaluation across runs. Second, *quantitative diagnostics* enable precise repairs (*e.g.*, "rotate $-2.3°$" rather than "fix alignment"), explaining the targeted convergence observed in ablations (Table 6). Third, *code localization* identifies specific lines needing modification, reducing cascading errors. Fourth, *binary rewards* from pass/fail outcomes provide clean training signals for reinforcement learning with verifiable rewards (RLVR), opening paths to self-improvement beyond prompting alone.

## A.2. System Prompt

The system prompt defines the agent's role, output format, workflow, and available tools. It constrains the agent to produce a single entry point (`main.scad` for OpenSCAD, `main.py` for Python CAD), enforces code quality requirements (named parameters with units, modular structure, explicit tolerances), and establishes a constraint checklist mechanism for tracking progress.

---

**System Prompt**

```
You are SPADA, an expert CAD coding assistant that can interact with a computer to solve tasks in
OpenSCAD, CadQuery, or build123d.
<ROLE>
- Produce clean, parametric, manufacturable CAD models in the requested language.
- Preserve the user's intent; do not add unrelated constraints.
- Prefer millimeter units, named parameters, and reusable modules.
<DELIVERABLE>
- main.scad is the entry point for OpenSCAD tasks; main.py is the entry point for CadQuery/build123d
tasks.
- Return only the code for the requested entry point.
<WORKFLOW>
1. Extract requirements (and spec if needed).
2. Write the entry point (main.scad or main.py).
3. Use inspect to validate geometry.
4. Use verifier to write and run tests (unless explicitly told to skip tests).
<TOOLING>
- terminal: explore files and run commands.
- file_editor: create or update files in the workspace.
- spec: extract a structured model spec from a prompt.
- inspect: compile the current model and report geometry stats.
- verifier: write and run pytest/SpecSCAD tests.
```

---

## A.3. Tool Interface and Implementation

The agent interacts with the execution environment through a fixed set of tools. Beyond core file editing and shell access, SPADA defines three CAD-specific tools that form the foundation of the test-driven workflow.

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

 5 active CAD communities: Thingiverse[5], Cults3D[6], CGTrader[7], MakerWorld[8], and Thangs[9];
- **GitHub Repos**: We uses Stack-V2-dedup as a starting point, which is a large-scale code corpus, we filter repositories that contain CAD programs in OpenSCAD, CadQuery, and Build123d by searching for characteristic file extensions and import statements. Besides, we also manually curated from awesome-series GitHub repositories that focus on code-based CAD designs;
- **Huggingface Datasets**: We incorporate existing code-based CAD datasets, including openscad;
- **Manually Selection**: We manually collect CAD programs from educational resources, tutorials, and example galleries to ensure coverage of fundamental design patterns.

**Preprocessing and Candidate Pool.**    Raw programs are normalized into a consistent workspace layout per CAD backend. We remove personally identifying strings (for example, emails and local user paths) using pattern matching. We remove near-duplicates using LSH-based matching over token shingles. We then execute each program in a unified sandbox with pinned dependencies, backend versions, and execution flags; only programs that run successfully and produce valid geometry artifacts are retained. The resulting executable pool contains 200K programs.

**Interface Contract Extraction.**    For each part task, we extract an interface contract $\Gamma$ that includes: (i) exported symbol name (module/function/class), (ii) parameter names and defaults, (iii) parameter types/units when available from the source language or annotations.

**Rewrite Description and Interface.**    In our early experiments, we found the data contamination issues when we directly use the original interfaces and related descriptions. Taking the similar method as GSM-Symbolic (Mirzadeh et al., 2024), we employ a rewriting LLM, specifically, gpt-oss-120b, to paraphrase the original task descriptions while preserving the core intent.

**Difficulty Control.**    Inspired by AutoCodebench (Chou et al., 2025), we use a moderately capable code model for filtering too easy tasks. Specifically, we use `gpt-oss-20b` for its high inference speed and coding ability. We let `gpt-oss-20b` generate code for all candidate tasks with a single attempt. If the generated code could pass all the interface tests, we consider this task as an easy task and filter it out.

## C.3. Test Categories

As code-based methods, similar to API definition with general SWE works, it includes interface definition and related parameters. Specifically for code-based parametric CAD, it further includes geometry validity and constraint satisfaction. In implementation, we group verifier tests into three categories:

---

[5]https://www.thingiverse.com/
[6]https://cults3d.com/
[7]https://www.cgtrader.com/
[8]https://makerworld.com/
[9]https://thangs.com/

| Constraint | Definition | Example |
|---|---|---|
| Point | The Point constraint is a frequently used constraint that minimizes the distance between two points. | |
| Axis | The Axis constraint minimizes the angle between two vectors. It is frequently used to align faces and control the rotation of an object. | |
| Plane | The Plane constraint is simply a combination of both the Point and Axis constraints. | |
| PointInPlane | PointInPlane positions the center of the first object within the plane defined by the second object. | |
| PointOnLine | PointOnLine positions the center of the first object on the line defined by the second object. | |
| FixedPoint | FixedPoint fixes the position of the given argument to be equal to the given point specified via the parameter of the constraint. | |
| FixedRotation | FixedRotation fixes the rotation of the given argument to be equal to the value specified via the parameter of the constraint. | |
| FixedAxis | FixedAxis fixes the orientation of the given argument's normal or tangent to be equal to the orientation of the vector specified via the parameter of the constraint. | |

*Figure 7.* **Assembly Constraint Examples.** In practice, those constraints could be combined to form complex assembly relations.

1. **Interface tests** ($\mathcal{T}_{\text{int}}$): Check that required APIs exist and that parameter names/defaults match the contract. We generate deterministic interface unit tests that import the entry point, check the signature, and instantiate with the provided configurations. In interface tests, it only checks whether the models fit the required interface and whether the models can be instantiated without syntax error.

2. **Geometry tests** ($\mathcal{T}_{\text{geo}}$): Check basic validity (*e.g.*, manifold status) and coarse properties (*e.g.*, bounding box). We set a threshold of $CD < 1 \times 10^{-3}$ between the generated model and the reference model to determine whether the geometry test passes (Guan et al., 2025).

3. **Constraint tests** ($\mathcal{T}_{\text{cons}}$): Check assembly relations (*e.g.*, mate/align), clearance bounds, and collision-free placement.

### C.4. Constraint Tests

For automatic constraint generation, we manually select multiple methods that fit the language's usage. In OpenSCAD, communities tend to use a built-in testing framework [10].

We implement deterministic assembly constraints over the executed assembly state $G$. Constraint types include: concentricity, coplanarity, parallelism, perpendicularity, distance/angle bounds, clearance bounds, and collision-free placement. Each constraint reports a structured failure record (constraint id, measured value, expected bound, and involved instances) to support targeted debugging.

In SPADA-Bench-Verified, the assembly constraints are one or both of these types.

### C.5. LLM-as-a-Judge Evaluation

For large-scale evaluation of dataset quality, following previous work (Govindarajan et al., 2026), we use LLM-as-a-judge to assess human-likeness of CAD code. We use `gpt-oss-20b` as the judge, which outputs integer scores from 0 to 5, where 5 indicates code that is most human-like.

---

[10]`https://github.com/openscad/openscad/issues/4075`

**LLM-as-a-Judge System Prompt**

```
You are a strict rater for whether a code snippet looks human-written.
Task: Given a CAD script, output a human-likeness score as an integer from 0 to 5, where 5 is most like
human.
Score meanings:
0: Almost surely an automatic primitive dump; no meaningful structure.
1: Mostly automatic; minor human touches but still dump-like.
2: Mixed; some structure but many dump signals.
3: Plausible human; structure exists but looks somewhat mechanical.
4: Clearly human-like; modular and readable with minor issues.
5: Strongly human-like; good abstraction, parameters, maintainable style.
What to look for (human-like signals):
- Parameterization: named variables for key dimensions.
- Decomposition: modules/functions for repeated parts.
- Control structure: loops for symmetry and repetition.
- Naming: meaningful identifiers; consistent terminology.
- Readability: consistent indentation; spacing; small blocks.
- Comments: helpful and not excessive.
- Maintainability: avoids long repeated blocks; uses helper modules.
Automatic / dump-like signals:
- Many repeated translate/rotate/cube lines with small numeric changes.
- "BLOCK N" markers or other auto-export comments.
- Many float literals with fixed formatting.
- Little or no variables, loops, modules, or reuse.
- Very high repetition of near-identical lines.
Output format requirements:
- Output ONLY valid JSON.
- No extra text outside JSON.
- JSON keys: score (int 0..5), rationale (string), confidence (float 0..1), signals (object).
```

**LLM-as-a-Judge User Prompt Template**

```
Code to rate:
--BEGIN CODE--
{code}
--END CODE--
Return ONLY the JSON object with keys: score, rationale, confidence, signals.
In signals, include short, concrete evidence such as: modules, assignments, loops, repetition patterns,
numeric-literal patterns, comments, formatting consistency, and any dump markers.
```

## C.6. Geometric Fidelity Metric Details

Both generated and ground truth CAD models are first exported to STL format, then converted to point clouds by uniformly sampling **2000 points** from mesh surfaces. We apply **ICP (Iterative Closest Point) alignment** (Besl & McKay, 1992) for optimal rigid transformation before computing distances.

**Chamfer Distance (CD)** measures bidirectional average nearest-neighbor distance between point clouds $P$ (generated) and $Q$ (ground truth):

$$\text{CD}(P,Q) = \frac{1}{2|P|} \sum_{p \in P} \min_{q \in Q} \|p - q\|_2 + \frac{1}{2|Q|} \sum_{q \in Q} \min_{p \in P} \|q - p\|_2 \tag{5}$$

**Hausdorff Distance (HDD)** measures worst-case geometric deviation:

$$\text{HDD}(P,Q) = \max \left\{ \sup_{p \in P} \inf_{q \in Q} \|p - q\|_2, \sup_{q \in Q} \inf_{p \in P} \|q - p\|_2 \right\} \tag{6}$$