# OpenReview forum: "SPADA: A Verifiable Test-Driven Agent for Controllable Parametric CAD Assembly Generation"
_ICML.cc/2026/Conference — ICML 2026 regular_

### Official Review · Reviewer_Qm5u · 2026-03-02

**Soundness:** 2
**Presentation:** 3
**Significance:** 3
**Originality:** 3
**Overall Recommendation:** 4
**Confidence:** 4

**Summary:**

The paper introduces SPADA, a test-driven agent for generating controllable, parametric CAD assemblies as executable code, coupled with deterministic verification tests that enforce geometric and relational constraints. It also presents SPADA-Bench-Verified, a large, multi-language benchmark (OpenSCAD, CadQuery, Build123d) with human-verified, deterministic verifiers for both parts and assemblies. Experiments across parts and assemblies demonstrate significant gains in validity, geometric fidelity, and constraint satisfaction over zero-shot LLM/VLM baselines and recent code-based CAD systems, and ablations indicate that the verifier-driven loop is the key driver of improvements.

**Compliance With Llm Reviewing Policy:**

Affirmed.

**Final Justification:**

I will keep my positive review.

**Key Questions For Authors:**

1.	The paper lacks an assessment of whether the generated test scripts are accurate and reliable for verification. Conducted human audits to verify the correctness and coverage of these generated test files would significantly enhance the soundness of the proposed framework.
2.	Could you provide per-language performance results or breakdowns by task difficulty (e.g., categorized by the number of parts or the type/number of constraints)? It is crucial to understand whether the conclusions hold consistently across different CAD ecosystems and complexity levels.
3.	To isolate the contribution of the SPADA agent logic from the benefits of iterative refinement, the authors should include a comparison where the baselines are also equipped with a similar verification/repair loop.
4.	Under what specific circumstances does the agent typically fail or reach the maximum iteration limit? Adding a dedicated section for Failure Case Analysis would provide valuable insights into the current limitations of the model and help future research better follow and build upon your work.

**Limitations:**

yes

**Strengths And Weaknesses:**

Strengths:

1.	Proposes SPADA-Bench-Verified, a high-quality dataset covering three CAD ecosystems (OpenSCAD, CadQuery, Build123d) that includes deterministic verification code.

2.	Introduces an iterative verification-in-the-loop framework (SPADA) that substantially boosts the success rate of parametric code generation for both parts and assemblies.

3.	Evaluates on both parts and assemblies with clear validity (IR), geometry (IoU, CD), and constraint metrics (PR, APR), and provides an ablation isolating tool contributions, showing the verifier’s decisive impact.


Weaknesses:

1.	The evaluation primarily focuses on code-level execution. It remains unclear whether the geometric and topological constraints of the resultant 3D models are fully captured beyond the unit tests.

2.	There is no reported human audit to verify the soundness of the test code generated by the verifier. If the verification scripts themselves contain errors, the reported success rates may be misleading.

3.	It appears that the baselines were evaluated without the benefit of the same verifier-driven iterative loop. It is unclear whether the performance gain is due to the SPADA architecture itself or simply because the baselines were denied comparable compile/verify/repair opportunities.

4.	Missing references to the Appendix. Specifically, in Section 3.3, Section 3.4, Table 2, and Section 5.1, citations appear as placeholders (e.g., "Appendix ??").

---

> ### Author Rebuttal · Authors · 2026-03-31
>
> ## Test Quality & Human Audits
>
> We thank the reviewer for the concern on test coverage and soundness. SPADA-Bench-Verified ensures test correctness through **deterministic generation** for interface and geometry tests, and **LLM-assisted generation with expert verification** for constraint tests.
>
> 1. **Test Coverage.** SPADA-Bench-Verified verifies geometric and topological properties through three test categories **covering interface compliance, geometry validity, and assembly constraint satisfaction.** The constraint tests build on eight primitive constraint types (Appendix C.3, C.4) that compose into higher-level checks (e.g., collision-free placement). We acknowledge that some implicit properties (e.g., surface continuity) may not be fully captured and have discussed them in limitations (P8).
>
> 2. **Verifier Correctness.** Interface tests are derived via AST parsing of the reference signatures; geometry tests check manifold validity and CD against a fixed threshold. Both are generated by deterministic scripts whose correctness is guaranteed by construction. Constraint tests use an LLM-driven pipeline that integrates heterogeneous sources (e.g., intent text, code) upon a fixed template. A CAD expert verified the template, and we ran pilot experiments to validate correctness before scaling up.
>
> To address your concern, we **conducted a human audit focusing on constraint tests** with 200 tasks (100 parts, 100 assembly). Although LLM-based generation still under-constrains some complex models, it mostly generated correct, intended constraint tests, with a **98.5% accuracy on all sampled tests**. The observed failure patterns include overly strict tolerances and missing edge-case constraints for complex inter-part relationships. We corrected all identified errors and will include the results in the revised version.
>
> ## Baseline Comparisons
>
> We thank the reviewer for raising the question of isolating SPADA's contribution from general self-refinement. As requested, we ran this experiment on a subset of SPADA-Bench-Verified (100 parts, 100 assemblies). Self-refinement baselines (e.g., MEDA, CADCodeVerify) are already in our main comparisons (Table 3) and therefore excluded from this run. We compared with the following self-refinement baselines:
>
> **1. Fine-tuning-based methods.** CAD-Coder is a representative fine-tuned VLM that generates CadQuery code, also included in our main comparison (Table 3). It uses LLaVA 1.5 as backbone, which does not support tool-use as in SPADA.  We implemented the baseline instead following Reflexion [1], where the model generates code, receives execution and visual feedback, and produces a refinement autoregressively.
>
> **2. Zero-shot LLMs.**  A basic LLM self-refinement baseline is already in our ablation (Table 6), where `+ general tools` enables iterative refinement and `+ inspect` provides geometric feedback, which closely resembles standard self-refinement settings. We ran Gemini-3-flash w/Reflexion, and SPADA (w/o verifier) to decouple test-based feedback.
>
> As the table shown, **the performance gain is attributable to SPADA's architecture**. Reflexion alone yields limited gains for CAD-Coder (+3.5 APR). On the same backbone, SPADA (w/o verifier) reaches 45.0 vs. 27.5 APR (+17.5), showing that SPADA's structured tools contribute beyond standard self-refinement. Adding the verifier provides a further +8.0 APR, confirming that test-driven feedback are significantly effective.
>
> | Method | IR (%) ↓ | PR (%) ↑ | APR (%) ↑ |
> | --- | --- | --- | --- |
> | CAD-Coder | 13.5 | 17.8 | 9.5 |
> | CAD-Coder w/Reflexion | 10.0 | 22.5 | 13.0 |
> | Gemini-3-flash w/Reflexion | 5.0 | 44.2 | 27.5 |
> | SPADA (w/o `verifier`) | 2.5 | 54.2 | 45.0 |
> | **SPADA** | **1.5** | **66.0** | **53.0** |
>
> ## Fine-grained Analysis
>
> 1. **Per-language/difficulty and Breakdown.** Performance varies consistently across CAD languages and constraint complexity, with OpenSCAD achieving the highest scores and APR degrading as the number of constraints increases. We provide per-language and per-difficulty breakdowns Table here: https://anonymous.4open.science/r/spada-paper-EF25/tables-Qm5u.pdf. and analyze in detail in the revised appendix.
>
> 2. **Failure analysis.** The main failure patterns, which are also the primary cause of iteration-limit exhaustion, are menntioned in Section 5.4. These same patterns happened where the **agent repeatedly attempts repairs that do not address the root cause** (e.g., applying partial spatial fixes that do not converge), ultimately exhausting the iteration budget without passing all tests. We will add a detailed failure analysis section with representative examples in the revised appendix.
>
> ## Writing Issues
>
> We thank the reviewer for the detailed feedback on writing quality. Broken references and overall presentation will be improved in the revision for clarity.
>
> **Reference**
>
> [1] Shinn, N. et al. Reflexion: Language Agents with Verbal Reinforcement Learning. NeurIPS 2023.

---

> > ### Author Rebuttal · Reviewer_Qm5u · 2026-04-03
> >
> > Thank you for your response. It has addressed my concern, and I would therefore like to keep my original score.

---

> > > ### Author Response · Authors · 2026-04-04
> > >
> > > Thank you for taking the time to review our responses. We're pleased to hear that our clarifications addressed your concerns, and we truly appreciate your recommendation for acceptance.

---

### Official Review · Reviewer_TBFd · 2026-03-09

**Soundness:** 2
**Presentation:** 1
**Significance:** 3
**Originality:** 2
**Overall Recommendation:** 4
**Confidence:** 2

**Summary:**

This paper primarily focuses on CAD generation and establishes a benchmark comprising approximately 11.7K samples, supporting 3 CAD modalities. The proposed self-testing agent is model-agnostic and does not involve SFT. It is essentially about prompting, tools, and zero-shot learning. A key feature of this agent is its ability to automatically verify and test the code generated by LLM.

**Compliance With Llm Reviewing Policy:**

Affirmed.

**Final Justification:**

Please consider incorporating the rebuttal into the final version. I have raised my score.

**Key Questions For Authors:**

- What is the most valuable and significant tool that's used?
- The paper has too many obvious writing errors, such as many Appendix ?
- The writing of the paper is very redundant, e.g., right column 131-160.
- What is the API cost for the inference?

**Limitations:**

yes

**Strengths And Weaknesses:**

Strengths:
- The dataset size is good, spanning 3 major code-based CAD modalitis.
- results are compared on CADPrompt.
- the agent framework achieved good results on the proposed datasets.
- work is engineering sound.

Weaknesses:
- my biggest concern is that this benchmark seems very exclusive and tool-centric, only usable for various **internal tool calls**. In other words, even if the model is strong, it cannot directly use this benchmark simply because of *lack of environment*.
- This paper does not use SFT and cannot further fine-tune the model, largely due to the limitations of the weakness above.
- Compared to DeepCAD, it seems this benchmark only additionally has a *code with tests* mechanism, but its innovation is limited.
- CADPrompt is essentially also a feedback-based approach, and this paper uses code with tests. in my opinion, the only difference being more CAD-related **tools**. I feel its contribution to the AI community is limited.
- Do agents with many tools really count for innovation, especially when the tools are very exclusive 3D CAD tools, rather than general tools?

---

> ### Author Rebuttal · Authors · 2026-03-31
>
> ## Benchmark Accessibility & Novelty
>
> We would like to clarify that **SPADA-Bench-Verified is open-source, SFT-compatible, and distinct from DeepCAD in that it couples generation with assembly-level constraint verification**.
>
> **1. SPADA-Bench-Verified is built on open-source CAD backends** (e.g., OpenSCAD). No proprietary software (e.g., SolidWorks) is required. A reproducible environment is provided in Appendix B so anyone can replicate it with a single Docker build. Moreover, tool-use is now a standard capability across frontier closed and open models, so any capable model can use the benchmark directly. We would open-source after the paper is accepted.
>
> **2. SFT is compatible with SPADA-bench-Verified**, which is supported by provided **structured (intent, workspace) pairs** from the benchmark. We excluded SFT from the main paper to ensure the reported gains reflect the test-driven paradigm, not model-specific fine-tuning. To further address your concern, we conducted a 16-bit LoRA SFT with Qwen3-VL-30B-A3B, dividing SPADA-Bench-Verified into an SFT and a separate evaluation set of 200 tasks. As Table shown, while SFT improves one-pass generation, especially on IR, **the dominant gains come from SPADA's executable feedback loop**, proving that the benchmark supports further model improvement through both SFT and test-driven paradigms.
>
> | Method | IR (%) ↓ | PR (%) ↑ | APR (%) ↑ |
> | --- | --- | --- | --- |
> | Zero-shot | 35.0 | 9.4 | 3.5 |
> | SFT | 16.5 | 14.9 | 6.5 |
> | SPADA (Qwen) | 13.0 | 33.4 | 19.5 |
> | SPADA (Qwen w/SFT) | 11.5 | 34.8 | 20.5 |
>
> **3. SPADA-Bench-Verified goes beyond adding tests, coupling generation with programmatic assembly-level constraint verification that DeepCAD cannot express.**
>
> (a) DeepCAD addresses **unconditional single-part generation** from Sketch-Extrude (SE) sequences, evaluated by geometric similarity, while SPADA-Bench-Verified focuses on **text-conditioned generation with verifiable constraints**, where the output must satisfy various considerations which geometric metrics would not covered. Code is the most natural representation for verifying assembly-level constraints (e.g., spatial relations), since it exposes features that enable automatic testing. The SE representation used in DeepCAD **does not support this level of verification**.
>
> (b) SPADA-Bench-Verified probes **spatial reasoning and multimodal generation with constraints** through executable evaluation, which is important for LLM agents operating in multiple domains (e.g., engineering) [1]. Current frontier models produce syntactically valid code yet fail substantially on spatial and relational constraints (Table 3-4), a gap that was previously **rarely seen to be measurable** with existing benchmarks, including DeepCAD.
>
> ## SPADA Tools & Novelty
>
> We respectfully note that SPADA differs substantively from CADCodeVerify (method underlying CADPrompt), and clarify the difference, tool design, and cost below.
>
> **1. Compared with CADCodeVerify, SPADA is distinct through precise grounding and handling complex dependencies.** SPADA uses executable tests to drive the repair loop, which provided more precise and less ambiguous feedback than VQA-based feedback used in CADCodeVerify (Tables 3–4), since it provides **localized, fine-grained failure signals** for targeted correction. **The most valuable tool is `verifier`**, which is responsible for the largest single gain in the ablation (+8.2 APR, Table 6), and no equivalent tool exists in the self-refinement baselines. Furthermore, these signals can also serve as binary rewards, making SPADA promising for further improvement with RL-based methods (e.g., RLVR). The novelty lies in **the nature of the feedback each tool produces**, a design principle that is central to effective agent systems across domains.
>
> **2. SPADA uses only 5 tools built upon open-source backends (3 CAD-specific, 2 general), which is comparable or less than other CAD agents**: CAD-Assistant [2] uses 7 (5 specific, 2 general), CADDesigner [3] uses 6 (4 specific, 2 general). We agree that fewer, well-designed tools are preferable to many, and this is precisely why SPADA minimizes tool count to keep agents focused (Appendix A).
>
> **3. API cost.** Cost varies with iteration count (~4 iters/task avg). Gemini-3-flash is ~0.03–0.15 USD/task, GPT-5 is ~0.10–0.50 USD/task, and Qwen3-VL-235B is ~0.02–0.08 USD/task. Detailed analysis will be included in the revised Appendix.
>
> ## Writing Issues
>
> We thank the reviewer for the detailed feedback on writing quality. Broken references, redundancy in Section 3, and overall presentation will be improved in the revision for clarity.
>
> **References**
>
> [1] Qiu, Z. et al. Can large language models understand symbolic graphics programs? ICLR 2025.
>
> [2] Mallis, D. et al. CAD-Assistant: Tool-augmented VLLMs as generic CAD task solvers. ICCV 2025.
>
> [3] Fan, F. et al. CADDesigner: Conceptual design of CAD models based on general-purpose agent. arXiv:2508.01031.

---

> > ### Author Rebuttal · Reviewer_TBFd · 2026-04-02
> >
> > > The most valuable tool is verifier.
> >
> > In ML field (not in engineering field), what's the distinct difference between this `verifier` and *code verifier that is used to do the code (e.g., syntax) check*? because the authors claimed that it can provide localized, fine-grained failure signals.
> >
> > > making SPADA promising for further improvement with RL-based methods (e.g., RLVR).
> >
> > And because the authors use the wording *verifier*, it's good to see the results with RL-based methods.

---

> > > ### Author Response · Authors · 2026-04-04
> > >
> > > ## Verifier vs. Syntax Checker
> > >
> > > We appreciate the opportunity to clarify this distinction. The `verifier` performs **semantic verification over executed geometric outputs**, fundamentally different from a syntax checker. A syntax checker operates on source code and passes as long as the code compiles without runtime errors. The `verifier`, by contrast, operates on the 3D artifacts produced *after* successful compilation and checks whether those artifacts satisfy the intended interface, geometry, and assembly constraints. A syntax checker is comparable to static linting, whereas the `verifier` functions as runtime unit tests that assert correctness of program outputs. Code that compiles cleanly can still fail all three categories of verifier tests.
> > >
> > > **Interface tests** verify that the generated module exposes the required parameter signature with correct types and defaults; a renamed or missing parameter compiles but fails the contract. **Geometry tests** inspect executed output for manifold validity and bounding-box dimensions; syntactically correct code can produce degenerate geometry. **Constraint tests** compute spatial relations between parts after execution, e.g., reporting `pitch diameter = 22mm, expected 24mm` or detecting a 0.3mm collision. None of these are detectable from syntax checks, which is why the `verifier` contributes the largest single gain (+8.2 APR) in our ablation (Table 6).
> > >
> > > ## RL-based Methods
> > >
> > > We agree that RL results would strengthen the paper. As requested, we conducted an offline RL experiment and an online-RL probe to illustrate SPADA's potential for further improvement.
> > >
> > > **1. Offline RL.** We ran DPO [1], an widely adopted offline RL method that directly uses the preference as a reward function. For each training task, we sampled one-pass candidates from the SFT checkpoint (Qwen3-VL-30B-A3B, no agent loop), executed the benchmark's golden test suite against each, labeled passing candidates as chosen and failing ones as rejected, and trained on these preference pairs with the DPO loss.
> > >
> > > **2. Online RL probe.** A full online RL setup (e.g., GRPO) requires integrating the sandbox into the on-policy rollout loop, where each sample involves code generation, sandbox compilation, and full test-suite execution, with per-rollout latency of ~30-240 seconds. Existing CAD works using online RL rely on multi-GPU infrastructure and multi-day runs [2, 3], which exceeds the scope of the rebuttal period. Instead, we ran a **verifier-filtered post-training experiment as a preliminary offline probe toward online RL**. Recent work [4] shows that rejection sampling fine-tuning (RFT) with oracle-filtered data serves as a practical proxy for measuring reward signal quality before committing to full online RL. We used the golden tests' pass/fail outcome to filter training data, with the same 200-task evaluation split as our SFT experiment. We sampled 8 candidates per training task from the same SFT checkpoint, retained only candidates passing all golden tests, and fine-tuned on this filtered set.
> > >
> > > | Method | IR (%) ↓ | PR (%) ↑ | APR (%) ↑ |
> > > |---|---|---|---|
> > > | Zero-shot | 35.0 | 9.4 | 3.5 |
> > > | SFT | 16.5 | 14.9 | 6.5 |
> > > | SFT + RFT | 14.0 | 18.7 | 9.5 |
> > > | SFT + DPO | **13.0** | **20.2** | **11.0** |
> > > | SPADA (Qwen w/ SFT) | 11.5 | 34.8 | 20.5 |
> > > | SPADA (Qwen w/ RFT) | 10.0 | 36.2 | 22.0 |
> > > | SPADA (Qwen w/ DPO) | **9.5** | **37.1** | **23.0** |
> > >
> > > As the result shown, the verifier-filtered model (RFT) improves over plain SFT across all metrics, confirming that the verifier's signal provides discriminative supervision. DPO yields a further gain, showing that preference-based learning over verifier outcomes is effective. When combined with SPADA's compile-test-repair loop, APR reaches 22.0 (RFT) and 23.0 (DPO). The marginal gain on top of SPADA (20.5 to 22.0/23.0) is smaller, which is expected because SPADA's test-time loop already corrects many of the same errors that verifier-filtered training addresses. This is consistent with training-time and test-time improvements targeting overlapping error modes while remaining complementary. The result supports that SPADA-Bench-Verified is a suitable substrate for future reward-based training, especially for constraint-level properties difficult to capture with geometric similarity rewards. A full online RL experiment remains a clear future direction.
> > >
> > > **References**
> > >
> > > [1] Rafailov, R. et al. Direct preference optimization: Your language model is secretly a reward model. NeurIPS 2023.
> > >
> > > [2] Li, J. et al. ReCAD: Reinforcement learning enhanced parametric CAD model generation with vision-language models. AAAI 2026.
> > >
> > > [3] Kolodiazhnyi, M. et al. Cadrille: Multi-modal CAD reconstruction with online reinforcement learning. ICLR 2026.
> > >
> > > [4] Yan, Y. et al. VerifyBench: Benchmarking reference-based reward systems for large language models. ICLR 2026.

---

### Official Review · Reviewer_J8Rz · 2026-03-13

**Soundness:** 2
**Presentation:** 2
**Significance:** 3
**Originality:** 3
**Overall Recommendation:** 4
**Confidence:** 3

**Summary:**

This paper introduces SPADA, a test-driven agent specifically designed to address the challenges existing CAD generative models face in creating complex multi-part assemblies that are both parametrically editable and strictly compliant with physical constraints.

SPADA automatically synthesizes verification test code alongside the CAD assembly code, treating it as an "executable contract." Through a multimodal feedback loop of "compile-test-repair," the agent iteratively corrects geometric and logical errors. Coupled with the authors' newly released rigorous evaluation benchmark, SPADA-Bench-Verified, experimental results demonstrate that this method significantly outperforms existing baselines, reliably generating high-fidelity, parametric CAD assemblies that meet real-world engineering constraints.

**Compliance With Llm Reviewing Policy:**

Affirmed.

**Final Justification:**

The authors have provided additional experimental visualization results and real-world user studies, which I believe better showcase the paper's contributions. The authors have also committed to addressing the writing issues; therefore, I have decided to maintain my score of 4.

**Key Questions For Authors:**

Could you provide user studies conducted in real-world scenarios to further demonstrate the superior assembly results of the proposed method? Additionally, please discuss whether a user study involving domain experts is necessary for this specific task.

**Limitations:**

Yes

**Strengths And Weaknesses:**

Strengths:
1. Proposes an innovative compile–test–repair framework that simultaneously ensures geometric fidelity and constraint satisfaction.
2. Constructs a dedicated dataset specifically designed to evaluate complex challenges in CAD assembly.
3. Achieves SOTA results, significantly outperforming existing methodologies.

Weakness:
1. Experimentally, in addition to using LLMs as judges, I would like to see some real-world user studies and more visualization results.
2. Please include your supplementary materials as an appendix at the end of the main text; otherwise, the citations referring to the appendix within your manuscript will be problematic—for example, at lines 200, 206, 238, and 307.

---

> ### Author Rebuttal · Authors · 2026-03-31
>
> ## Real-world User Studies
>
> Thank you for raising this point. We agree that human evaluation is valuable for CAD assembly generation. Following common practice, we conducted a **blinded pairwise preference study** with a protocol similar to LMArena [1].
>
> We randomly sampled **100 assembly tasks** from SPADA-Bench-Verified and recruited **five human experts**, each familiar with Code-based CAD and mechanical engineering. For each task, experts were shown two anonymized, randomly ordered assembly outputs with implemented workspaces, with no method labels visible. Each expert compared pairs of generated assembly code across all method pairs and selected the preferred one based on constraint satisfaction, structural correctness, and parametric editability, yielding 796 pairwise judgments in total. To reduce positional bias, presentation order was randomized per comparison. Preferences were then aggregated using the Bradley-Terry model, where each method's latent strength score was estimated via maximum likelihood from all pairwise outcomes, and confidence intervals were computed via bootstrap resampling.
>
> We compared SPADA against the zero-shot baseline (Gemini-3-flash) and code-based CAD baselines with self-refinement (CADCodeVerify, MEDA). **SPADA achieves the highest Arena Score with a clear margin** over the next-best method, confirming that verifier-guided constraint satisfaction translates to a meaningful preference advantage among domain experts.
>
> | Rank | Method | Arena Score | Votes |
> |------|--------|-------|-------|
> | 1 | SPADA (Ours) | 1147.5±22.4 | 202 |
> | 2 | CADCodeVerify | 1089.8±24.7 | 200 |
> | 3 | Gemini-3-flash | 1021.3±26.1 | 198 |
> | 4 | MEDA | 943.2±28.3 | 196 |
>
> **Domain expert involvement is necessary** for this task. Assessing parametric assemblies requires judging mating correctness, collision avoidance, dimensional consistency, and other engineering constraints that general users may overlook even when an output looks visually plausible. Within the current submission, we therefore focus on objectively verifiable assembly correctness together with targeted expert evaluation. As our method supports Text2CAD, **users without CAD experience can also interact with SPADA through natural language descriptions or images**, which lowers the barrier to parametric CAD generation and makes a broader user study a promising direction for future work.
>
> ## More Visualization Results
>
> We will add a dedicated visualization section in the appendix focusing on SPADA's generation outputs in the revised manuscript. **Extended visualization results** are available here: https://anonymous.4open.science/r/spada-paper-EF25/figures-J8Rz.pdf.
>
> As Figures 1-4 show, SPADA **generates models across diverse categories**, including mechanical components (e.g., LED strip mounting clips, shaft coupler adapters, timing pulleys), functional objects (e.g., board game card holders, honeycomb pencil holders, laboratory vial racks), and everyday items (e.g., magnetic picture frames, heart lattice bowls, wall-mounted ball hooks).
>
> Figure 5 demonstrates SPADA's support for **downstream editing tasks**. Starting from an initial generation, users can issue natural-language edit commands (e.g., "extend the vertical arm and deepen the brace") and SPADA produces modified geometry that preserves the original design intent while satisfying updated constraints. We also include more detailed examples to illustrate SPADA's support for **multiple Code-based CAD backends** (Figures 6-9), showing broad generation diversity across OpenSCAD, CadQuery, and Build123d.
>
> ## Writing Issues
>
> Thank you for identifying the appendix reference issues. We will **correct all unresolved "Appendix ??" references** and properly integrate the supplementary material into the main document in the revised paper.
>
> **References**
>
> [1] Chiang, W.-L., et al. Chatbot Arena: An Open Platform for Evaluating LLMs by Human Preference. ICML 2024.

---

> > ### Author Rebuttal · Reviewer_J8Rz · 2026-04-02
> >
> > Thank you for your reply, which has solved my problem. I will keep my score.

---

> > > ### Author Response · Authors · 2026-04-04
> > >
> > > Thank you for taking the time to review our responses. We're pleased to hear that our clarifications addressed your concerns, and we truly appreciate your recommendation for acceptance.

---

### Official Review · Reviewer_mmdp · 2026-03-13

**Soundness:** 2
**Presentation:** 2
**Significance:** 3
**Originality:** 3
**Overall Recommendation:** 4
**Confidence:** 3

**Summary:**

This paper proposes SPADA, a verifier-guided agent framework for controllable parametric CAD assembly generation. The authors develop an iterative compile–test–repair loop where generated CAD code is executed and validated using deterministic tests that check geometric and assembly constraints. They also introduce SPADA-Bench-Verified, a benchmark for constraint-verifiable CAD tasks. Experiments show that SPADA significantly improves constraint satisfaction and assembly correctness compared with strong LLM baselines.

**Compliance With Llm Reviewing Policy:**

Affirmed.

**Final Justification:**

Thanks for the authors' rebuttal. I keep my original scores unchanged.

**Key Questions For Authors:**

Can the verifier framework generalize to other CAD representations or CAD software ecosystems?

**Limitations:**

The current framework focuses on code-based CAD generation with explicit constraint verification. Its effectiveness for more complex assemblies or other CAD representations remains to be explored.

**Strengths And Weaknesses:**

++
The verifier-driven loop is a practical way to enforce engineering constraints in CAD generation.
The proposed benchmark introduces executable constraint verification rather than only geometric similarity.
Experimental results show clear improvements over strong baselines.

--
The framework relies heavily on predefined verifiers and test generation, which may limit scalability.
Some improvements may come from the engineering pipeline rather than fundamentally new modeling ideas.

---

> ### Author Rebuttal · Authors · 2026-03-31
>
> ## Scalability of the Predefined Verifier
>
> We thank the reviewer for this important point and agree that verifier scalability is central to the practicality of our framework. Within the current scope of our work, **the framework, including the `verifier` we used, is already scalable**. We address scalability for each component separately, including SPADA-Bench-Verified (the proposed benchmark) and SPADA (the proposed method).
>
> For SPADA-Bench-Verified, the **construction pipeline (Figure 6) already scales verifier generation** to 11.7k tasks through LLM-assisted test generation from structured templates, followed by human verification for quality assurance. Because the three test categories ($T_{int}$, $T_{geo}$, $T_{cons}$) are defined over structured interfaces and geometric predicates, they are amenable to template-based automated generation. Extending the benchmark to new tasks or ecosystems therefore requires adding structured task metadata and executable code, from which the pipeline can generate verifiers with the same automated approach.
>
> For SPADA, the agent generates test code from the extracted specification at each iteration and uses execution feedback to guide correction during inference. Scaling SPADA to larger task volumes depends on the efficiency of the underlying execution infrastructure. Modern agent infrastructure also makes large-scale execution increasingly practical through isolated and parallel sandbox environments such as [Daytona](https://www.daytona.io/) and [E2B](https://e2b.dev/). Together, these infrastructure developments make scaling SPADA to larger task volumes practical without requiring architectural changes to the framework.
>
> ## Method Novelty Beyond the Self-Refinement Loop
>
> We partly agree that some improvement comes from the self-correcting loop, which is also implemented by other baseline methods. However, we respectfully note that our contribution is a **test-driven formulation for code-based parametric assembly generation** in which relational and geometric constraints become executable contracts, **going beyond a tool-specific engineering pipeline**. Within the current scope, the framework achieved state-of-the-art performance across three major code-based CAD ecosystems.
>
> To further address your concern, we ran an experiment on a subset of SPADA-Bench-Verified (100 parts, 100 assemblies) to decouple the effect of the self-refinement loop itself. We excluded the self-refinement baselines (e.g., MEDA, CADCodeVerify) from this run because they are already compared with SPADA in our main results (Table 3). Instead, we compared against Reflexion [1], one of the most widely adopted self-refinement methods, using Gemini-3-flash (backbone of SPADA) as the shared backbone, to clearly isolate the contribution of our test-driven formulation from the self-refinement loop itself.
>
> The results are as follows. **The performance gain is attributable to SPADA's test-driven formulation**, which includes structured specification extraction via `spec`, geometric inspection via `inspect`, and constraint-aware test generation. SPADA without the verifier already outperforms Reflexion by +17.5 APR, showing that the **test-driven formulation itself contributes substantially beyond generic verbal self-refinement**. Adding the `verifier` provides a further +8.0 APR gain through targeted, quantitative repair signals, confirming that the deterministic verification component is an additional and complementary source of improvement.
>
> | Method | IR (%) ↓ | PR (%) ↑ | APR (%) ↑ |
> | --- | --- | --- | --- |
> | Gemini-3-flash w/Reflexion | 5.0 | 44.2 | 27.5 |
> | SPADA (w/o `verifier`) | 2.5 | 54.2 | 45.0 |
> | **SPADA (Ours)** | **1.5** | **66.0** | **53.0** |
>
> ## Generalization to Other CAD Ecosystems
>
> We thank the reviewer for raising this question. Within the validated scope of this paper, SPADA **already generalizes across three code-based CAD ecosystems**. More broadly, we expect the verifier framework to **transfer to other CAD software ecosystems whenever geometry or assembly APIs are programmatically accessible**, because the same constraint-checking pattern can be implemented through executable tests in that environment. As a concrete example, CAD-Assistant [2] is built on FreeCAD, a different CAD ecosystem, and already implements constraint-checking tools. One could similarly have the agent write analogous tests within that ecosystem using the same executable-contract patterns.
>
> For other CAD representations, our current evidence supports code-based parametric representations with explicit programmatic execution. Generalization to non-code-based representations or proprietary CAD formats remains outside the validated scope of this work and is an important direction for future study.
>
> **Reference**
>
> [1] Shinn, N. et al. Reflexion: Language Agents with Verbal Reinforcement Learning. NeurIPS 2023.
>
> [2] Mallis, D. et al. CAD-Assistant: Tool-Augmented VLLMs as Generic CAD Task Solvers. ICCV 2025.

---

### Decision · Program_Chairs · 2026-04-30

**Decision:**

Accept (regular)

**Comment:**

The paper consistently received strong scores, and the rebuttal/discussion helped clarify some of the reviewers' concerns. The reviewers kept their scores, and the overall consensus is to accept the work.

This submission introduces SPADA as a test-driven agent developed to overcome the limitations of current CAD generative models in producing complex, multi-part assemblies that are both parametrically editable and adhere strictly to physical constraints.